# HCN channel-mediated neuromodulation can control action potential velocity and fidelity in central axons

Niklas Byczkowicz[1], Abdelmoneim Eshra[1], Jacqueline Montanaro[2], Andrea Trevisiol[3], Johannes Hirrlinger[1,3], Maarten HP Kole[4,5], Ryuichi Shigemoto[2], Stefan Hallermann[1]*

[1]Carl-Ludwig-Institute for Physiology, Medical Faculty, University Leipzig, Leipzig, Germany; [2]Institute of Science and Technology Austria (IST Austria), Klosterneuburg, Austria; [3]Department of Neurogenetics, Max-Planck-Institute for Experimental Medicine, Göttingen, Germany; [4]Department of Axonal Signaling, Netherlands Institute for Neuroscience, Royal Netherlands Academy of Arts and Sciences, Amsterdam, Netherlands; [5]Cell Biology, Faculty of Science, University of Utrecht, Padualaan, Netherlands

**Abstract** Hyperpolarization-activated cyclic-nucleotide-gated (HCN) channels control electrical rhythmicity and excitability in the heart and brain, but the function of HCN channels at the subcellular level in axons remains poorly understood. Here, we show that the action potential conduction velocity in both myelinated and unmyelinated central axons can be bidirectionally modulated by a HCN channel blocker, cyclic adenosine monophosphate (cAMP), and neuromodulators. Recordings from mouse cerebellar mossy fiber boutons show that HCN channels ensure reliable high-frequency firing and are strongly modulated by cAMP (EC$_{50}$ 40 μM; estimated endogenous cAMP concentration 13 μM). In addition, immunogold-electron microscopy revealed HCN2 as the dominating subunit in cerebellar mossy fibers. Computational modeling indicated that HCN2 channels control conduction velocity primarily by altering the resting membrane potential and are associated with significant metabolic costs. These results suggest that the cAMP-HCN pathway provides neuromodulators with an opportunity to finely tune energy consumption and temporal delays across axons in the brain.
DOI: https://doi.org/10.7554/eLife.42766.001

*For correspondence:
hallermann@medizin.uni-leipzig.de

**Competing interests:** The authors declare that no competing interests exist.

## Introduction

HCN channels are expressed in the heart and nervous system and comprise four members (HCN1–HCN4), which differ in their kinetics, voltage-dependence and degree of sensitivity to cyclic nucleotides such as cAMP (*Biel et al., 2009*; *Robinson and Siegelbaum, 2003*). Membrane hyperpolarization activates HCN channels and causes a depolarizing mixed sodium/potassium (Na$^+$/K$^+$) current. In the heart, the current through HCN channels ($I_f$) mediates the acceleratory effect of adrenaline on heart rate by direct binding of cAMP (*DiFrancesco, 2006*). In neurons, the current through HCN channels ($I_h$) controls a wide array of functions, such as rhythmic activity (*Pape and McCormick, 1989*) and excitability (*Tang and Trussell, 2015*). In addition to the somatic impact, HCN channels are expressed throughout various subcellular compartments of neurons (*Nusser, 2012*). For example, patch-clamp recordings from dendrites in pyramidal neurons have revealed particularly high densities of HCN channels that act to control the local resting potential and leak conductance, thereby playing important roles in regulating synaptic integration (*George et al., 2009*; *Harnett et al., 2015*; *Kole et al., 2006*; *Magee, 1999*; *Williams and Stuart, 2000*).

On the contrary, the expression and role of $I_h$ in the axon is less studied. $I_h$ seems to control the strength of synaptic transmission in the crayfish and the *Drosophila* neuromuscular junction (*Beaumont and Zucker, 2000*; *Cheung et al., 2006*). However, presynaptic recordings from the vertebrate calyx of Held in the auditory brainstem found $I_h$ to only have a marginal effect on neurotransmitter release (*Cuttle et al., 2001*), but to exert a strong influence on the resting membrane potential (*Cuttle et al., 2001*; *Kim and von Gersdorff, 2012*) and on vesicular neurotransmitter uptake (*Huang and Trussell, 2014*). At the synaptic terminals of pyramidal neurons in the cortex of mice, HCN channels inhibit glutamate release by suppressing the activity of T-type $Ca^{2+}$ channels (*Huang et al., 2011*).

Besides a potential impact on neurotransmitter release, axonal $I_h$ could play a role in the propagation of action potentials. Indeed, in axons of the stomatogastric nervous system of lobsters (*Marder and Bucher, 2001*), the action potential conduction was affected by dopamine acting via axonal HCN channels (*Ballo et al., 2010*; *Ballo et al., 2012*). In vertebrates, studies on action potential propagation by Waxman and coworkers indicated that $I_h$ counteracts the hyperpolarization of the membrane potential during periods of high-frequency firing (*Baker et al., 1987*; *Birch et al., 1991*; *Waxman et al., 1995*), and that it participates in ionic homeostasis at the node of Ranvier (*Waxman and Ritchie, 1993*). More recent investigations found $I_h$ to be crucial for the emergence of persistent action potential firing in axons of parvalbumin-positive interneurons (*Elgueta et al., 2015*), but $I_h$ seems to have an opposing effect on the excitability at the axon initial segment, where its activation reduces the probability of action potential initiation (*Ko et al., 2016*). Finally, there is evidence from extracellular recordings that blocking $I_h$ decreases the action potential conduction velocity in unmyelinated central axons (*Baginskas et al., 2009*; *Soleng et al., 2003*) and peripheral axons of vertebrates (*Grafe et al., 1997*). However, the neuromodulation of conduction velocity and the underlying cellular membrane mechanisms are not known in vertebrate axons.

Here, we demonstrate a decrease or increase in conduction velocity in central axons as a result of the application of HCN blockers or neuromodulators. To gain mechanistic insights into the modulation of conduction velocity by HCN channels, we performed recordings from *en passant* cerebellar mossy fiber boutons (cMFB; *Ritzau-Jost et al., 2014*; *Delvendahl et al., 2015*). We found that HCN channels in cMFBs mainly consist of the HCN2 subunit, are ~7% activated at resting membrane potential, ensure high-frequency firing, and control the passive membrane properties. Whole-cell and perforated patch clamp recordings from cMFBs demonstrated a strong dependence of HCN channels on intracellular cAMP concentration with an $EC_{50}$ of 40 µM and a high endogenous cAMP concentration of 13 µM. Computational modeling indicated that the resting membrane potential controls conduction velocity and that the activity of the HCN channel is metabolically expensive. These data reveal the existence of a mechanism to modulate conduction velocity bidirectionally in the central nervous system, which is shared among different types of axons.

## Results

### Bidirectional modulation of conduction velocity

To investigate whether HCNs affect conduction velocity, we recorded compound action potentials in three different types of axons (*Figure 1*). Application of the specific HCN channel blocker ZD7288 (30 µM) decreased the conduction velocity by 8.0 ± 2.8% in myelinated cerebellar mossy fibers (n = 14), by 9.2 ± 0.9% in unmyelinated cerebellar parallel fibers (n = 15), and by 4.0 ± 0.8% in optic nerves (n = 4; see *Figure 1* and its legend for statistical testing). As some studies implied that ZD7288 might have unspecific side effects, such as blocking voltage-dependent $Na^+$ channels (*Chevaleyre and Castillo, 2002*; *Wu et al., 2012*), we recorded $Na^+$ currents from 53 cMFBs and found no change in the amplitude or kinetics of voltage-dependent $Na^+$ currents after ZD7288 application (*Figure 1—figure supplement 1*),suggesting that under our conditions and at a concentration of 30 µM, ZD7288 did not affect the $Na^+$ currents. Because of the modulation of HCN channels by intracellular cAMP, we measured conduction velocity during the application of 8-bromoadenosine 3′,5′-cyclic monophosphate (8-Br-cAMP; 500 µM), a membrane-permeable cAMP-analog. The conduction velocity increased by 5.9 ± 2.8% in cerebellar mossy fibers (n = 17), by 3.7 ± 1.4% in parallel fibers (n = 10), and by 4.6 ± 0.6% in optic nerves (n = 5; see *Figure 1* and its legend for statistical

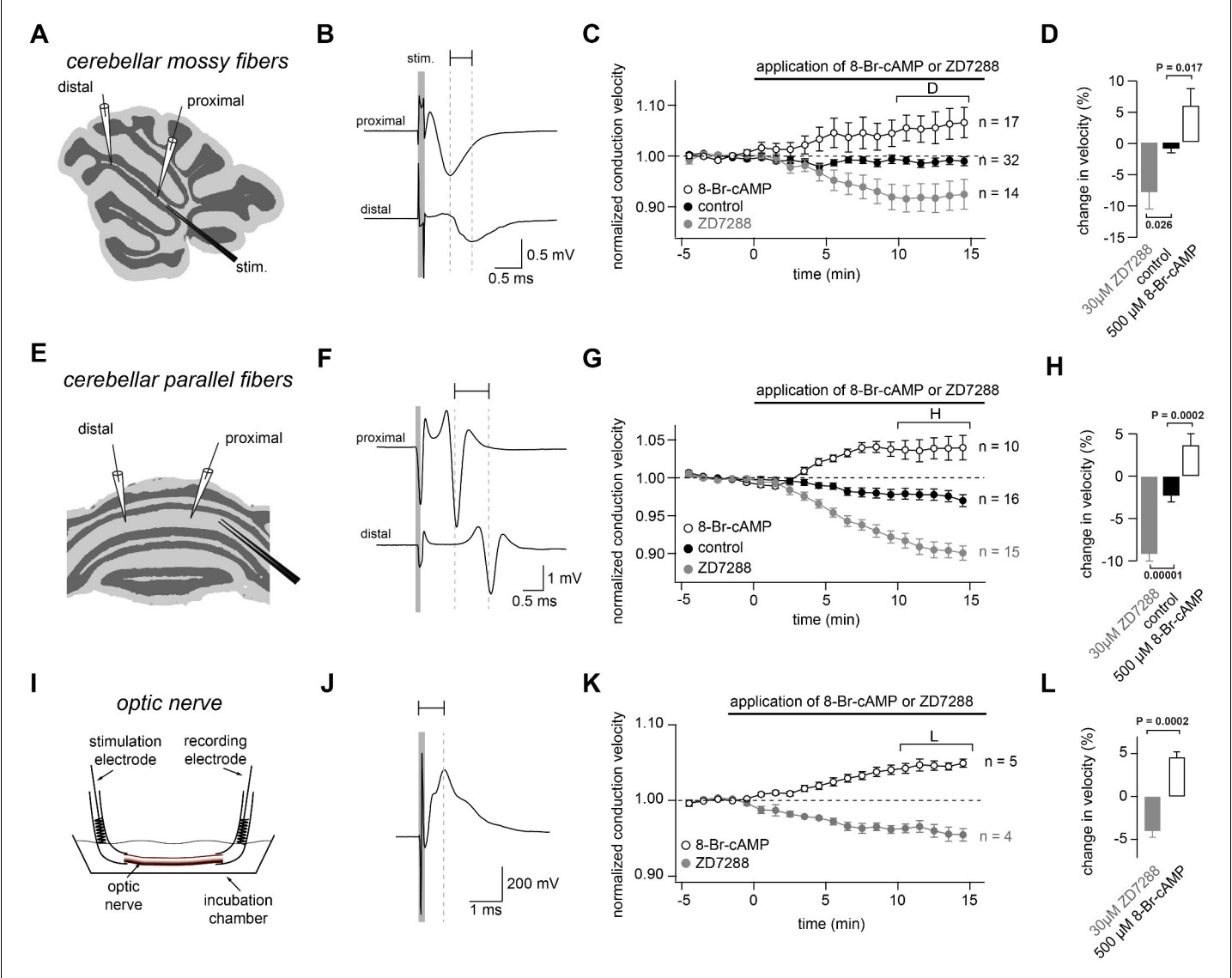

**Figure 1.** Bidirectional modulation of conduction velocity. (A) Recording configuration of conduction velocity in mossy fibers using a bipolar tungsten stimulation electrode (stim.) and two glass recording electrodes. (B) Example of compound action potentials recorded with two electrodes positioned at different distances in relation to the stimulation electrode. The stimulation (100 μs duration) is indicated by the gray bar. Each trace is an average of 50 individual compound action potentials recorded at 1 Hz. The delay between the peak of the proximal and the distal compound action potential is indicated by a horizontal line. (C) Average normalized mossy fiber conduction velocity, during bath application (starting at t = 0 min)of ZD7288 (30 μM) or 8-Br-cAMP (500 μM). (D) Average relative changes in conduction velocity of mossy fiber measured 10 to 15 min after beginning the application of ZD7288 or 8-Br-cAMP (bracket in C). $P_{ANOVA}$ = 0.00015. $P_{Kruskal-Wallis}$ = 0.00044. The individual P values of the Dunnett test for multiple comparisons with the control are indicated. (E) Schematic illustration of the experimental configuration used to record from cerebellar parallel fibers. (F) Examples of compound action potentials recorded from parallel fibers, as in panel (B). (G) Normalized conduction velocity in parallel fibers, as in panel (C). (H) Average relative changes in conduction velocity parallel fibers, as in panel (D). $P_{ANOVA}$ = $10^{-9}$. $P_{Kruskal-Wallis}$ = $10^{-8}$. The individual P values of the Dunnett test for multiple comparisons with the control are indicated. (I) Schematic illustration of the experimental configuration used to record from optic nerve. (J) Examples of compound action potentials recorded from optic nerve, as in panel (B). (K) Normalized conduction velocity in optic nerve, as in panel (C). (L) Average relative changes in conduction velocity of optic nerve, as in panel (D). $P_{T-Test}$ = 0.0002. $P_{Wilcoxon-Mann-Whitney-Test}$ = 0.004.
DOI: https://doi.org/10.7554/eLife.42766.002

The following figure supplement is available for figure 1:

**Figure supplement 1.** ZD7288 does not alter Na$^+$ currents in cMFBs.
DOI: https://doi.org/10.7554/eLife.42766.003

testing). These results indicate that HCN channels control the conduction velocity in both myelinated and unmyelinated central axons.

## Neuromodulators differentially regulate conduction velocity via HCN channels

To investigate a modulation of conduction velocity by physiological neuromodulators, we focused on the cerebellar parallel fibers, in which the velocity could be most accurately measured, and then applied several modulators known to act via cAMP-dependent pathways (*Figure 2*). Application of 200 µM norepinephrine (NE) resulted in a relatively fast increase in conduction velocity (1.9 ± 0.8%; n = 6; see *Figure 2A and C–E*; see legend for statistical testing), consistent with the existence of β-adrenergic receptors in the cerebellar cortex (*Nicholas et al., 1993*) that increase the cAMP concentration via $G_s$-proteins. On the other hand, the application of either 200 µM serotonin (–3.5 ± 0.5%; n = 11), 200 µM dopamine (–5.0 ± 0.7%; n = 13) or 200 µM adenosine (–7.2 ± 0.6%; n = 5) resulted in a continuous decrease of the conduction velocity (*Figure 2B and C–E*), consistent with the existence of $G_i$-coupled receptors for serotonin, dopamine, and adenosine in the molecular layer of the cerebellum (*Geurts et al., 2002*; *Schweighofer et al., 2004*), which decrease the cAMP concentration. Although we used high concentrations of the agonists and off-target effects cannot be excluded (e.g., the activation of dopamine receptors by NE [*Sánchez-Soto et al., 2016*]), these data nevertheless indicate that physiological neuromodulators can both increase and decrease action potential conduction velocity, depending on the type of neuromodulator and receptor.

In addition to HCN channels, some voltage-gated $Na^+$, $K^+$, and $Ca^{2+}$ channels can be modulated by the intracellular cAMP-pathway (*Burke et al., 2018*; *Yang et al., 2013*; *Yin et al., 2017*). To address the contribution of other channels on the neuromodulation of the conduction velocity, we performed a set of experiments in which HCN channels were first blocked by 30 µM ZD7288 before we applied three modulatory substances that had significantly increased or decreased conduction velocity in previous experiments. With ZD7288 continuously present in the recording solution, the conduction velocity of parallel fibers decreased over the course of 20 min (*Figure 2F*; see Materials and methods). Compared with control conditions (i.e. only ZD7288), adding 8Br-cAMP (500 µM), adenosine (200 µM) or NE (100 µM) at t = 0 min (i.e. 25 min after application of ZD7288) did not change the conduction velocity. The average conduction velocity between t = 10 and 15 min was decreased by –3.3 ± 2.4% for cAMP (n = 9), –4.6 ± 1.6% for adenosine (n = 9) and –3.7 ± 1.2% for NE (n = 7) when compared to the average velocity between t = 0 and 5 min in the baseline recording. This was not significantly different from the decrease measured in the presence of ZD7288 alone (control, –3.3 ± 1.4%; n = 7, see *Figure 2G*), indicating that the previously shown effects of cAMP and neuromodulators on conduction velocity are mainly mediated by HCN channels.

## Cerebellar mossy fiber terminals have a prominent voltage sag

To investigate the membrane and signaling mechanisms underlying the bidirectional control of conduction velocity, we focused on cerebellar mossy fibers, which allow whole-cell recordings with direct access to the cytoplasmic compartment (*Figure 3A*). Because of a long membrane length constant and the slow gating of HCN channels, recordings from *en passant* cMFBs are well suited for the investigation of the ionic basis of conduction velocity in adjacent axonal compartments. Injection of depolarizing currents during current-clamp recordings evoked a single action potential, while injection of hyperpolarizing currents generated a substantial 'sag' (*Figure 3B*; *Rancz et al., 2007*; *Ritzau-Jost et al., 2014*) (i.e. a delayed depolarization towards the resting potential, which is a hallmark of the presence of $I_h$) (*Biel et al., 2009*; *Robinson and Siegelbaum, 2003*). At a potential of, on average, –150 mV, the sag ratio (calculated from the peak and steady state amplitude as indicated in *Figure 3C* (*George et al., 2009*) was 0.497 ± 0.030 (n = 12).

## HCN channels support high-frequency action potential firing

Using direct recordings from cMFBs, we first aimed to investigate the impact of HCN channels on action potential firing. To this end, we analyzed action potentials elicited by current injections into the cMFBs (data not shown) as well as traveling action potentials elicited by axonal stimulation with a second pipette (*Figure 4A*). In both cases, the amplitude and half-duration of the action

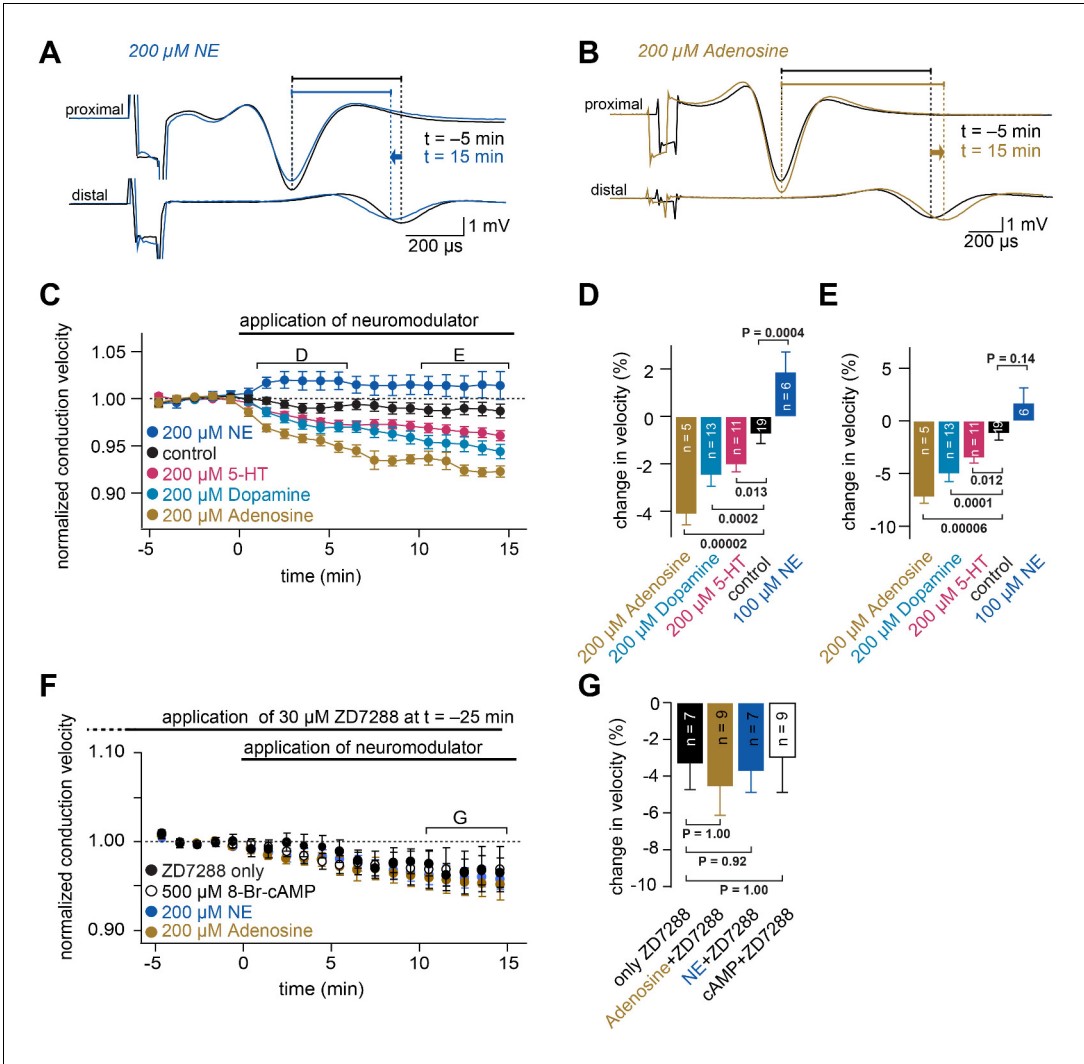

**Figure 2.** Neuromodulators differentially regulate conduction velocity via HCN channels. (**A**) Example of compound action potentials recorded in parallel fibers. Each trace is an average of signals recorded over a period of 1 min, before (at −5 min, black line) and after application of 200 μM NE (at 15 min, blue line). The delays between the peak of the proximal action potential and the distal compound action potential are indicated by horizontal bars. Traces are aligned to the peak of the compound action potentials recorded with the proximal electrode. (**B**) Example of compound action potentials as shown in panel (**A**) with the application of 200 μM adenosine. (**C**) Average normalized conduction velocity in cerebellar parallel fibers during the application of various neuromodulators that are known to act via cAMP-dependent pathways. (**D**) Average relative change in conduction velocity measured from 1 to 6 min after the start of neuromodulator application (bracket marked D in panel (**C**)). $P_{ANOVA} = 9*10^{-10}$. $P_{Kruskal-Wallis} = 3*10^{-8}$. The individual P values of the Dunnett test for multiple comparisons with the control are indicated. (**E**) Average relative change in conduction velocity measured from 10 to 15 min after the start of application of the neuromodulators (bracket marked E in panel (**C**)). $P_{ANOVA} = 3*10^{-7}$. $P_{Kruskal-Wallis} = 3*10^{-7}$. The individual P values of the Dunnett test for multiple comparisons with the control are indicated. (**F**) Average normalized conduction velocity in parallel fibers. 30 μM ZD7288 was applied 25 min before the start of the application of the neuromodulators. ZD7288 remained in the solution during recording to ensure continuously blocked HCN channels. At t = 0 min, 8-Br-cAMP, adenosine or NE was added to the solution. (**G**) Average relative change in conduction velocity measured 10 to 15 min after the start of application of the neuromodulators (bracket marked G in panel (**F**)). $P_{ANOVA} = 0.91$. $P_{Kruskal-Wallis} = 0.77$. The individual P values of the Dunnett test for multiple comparisons with the control are indicated.

DOI: https://doi.org/10.7554/eLife.42766.004

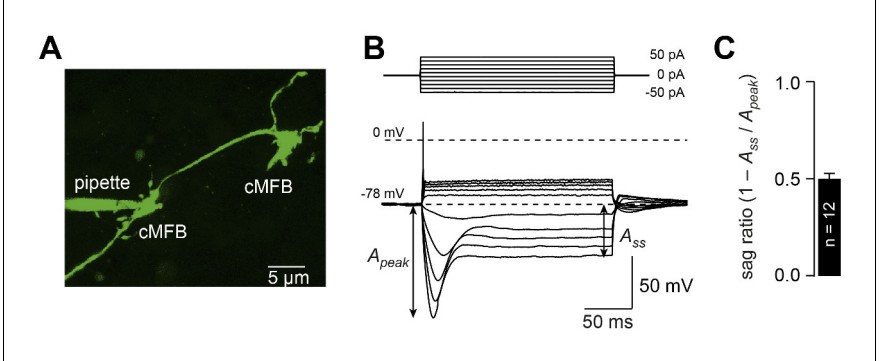

**Figure 3.** Cerebellar mossy fiber terminals have a prominent voltage sag. (A) Two-photon microscopic image of a whole-cell patch-clamp recording from a cMFB (green) filled with the fluorescence dye Atto 488 in an acute cerebellar brain slice of an adult 39-day-old mouse (maximal projection of stack of images). (B) Characteristic response of a cMFB to current injection: depolarizing pulses evoked a single action potential and hyperpolarizing pulses evoked a strong hyperpolarization with a sag. (C) Average sag ratio of 12 cMFB recordings.
DOI: https://doi.org/10.7554/eLife.42766.005

potentials elicited at 1 Hz were not significantly affected by the application of 30 µM ZD7288 (data not shown and *Figure 4B and C*, respectively), indicating that HCN channels do not alter the active membrane properties profoundly.

However, cerebellar mossy fibers can conduct trains of action potentials at frequencies exceeding 1 kHz (*Ritzau-Jost et al., 2014*), making them an ideal target to investigate the impact of axonal HCNs on the propagation of high-frequency action potentials. Blocking HCN channels significantly impaired the ability of mossy fibers to fire at high frequencies (20 stimuli at 200–1666 Hz). In the examples illustrated in *Figure 4D*, the failure-free trains of action potentials could be elicited at up to 1.1 kHz under control conditions and at up to 500 Hz when ZD7288 was present in the extracellular solution. The average failure-free frequency was reduced from $854 \pm 60$ Hz under control conditions to $426 \pm 63$ Hz in the presence of ZD7288 (n = 20 and 10, respectively; $P_{T\text{-Test}} = 0.0002$; *Figure 4E*). Action potential broadening and amplitude reduction was more pronounced in the presence of ZD7288. For example, during trains of action potentials at 200 Hz, the half-duration of the 20th action potential was $109.6\% \pm 1.5\%$ and $141.7\% \pm 7.0\%$ of the half-duration of the 1st action potential for control and ZD7288-treated MFBs, respectively (n = 20 and 10; $P_{T\text{-Test}} = 0.02$; *Figure 4E*). The amplitude of the 20th action potential was $96.5\% \pm 0.8\%$ and $82.9\% \pm 1.6\%$ of the 1st action potential for control and ZD7288-treated MFBs, respectively (n = 20 and 10; $P_{T\text{-Test}} = 0.02$; *Figure 4E*). Furthermore, the delay during trains of action potentials at 200 Hz increased by ~20% in the presence of ZD7288 but decreased by ~5% in control recordings (*Figure 4F*), indicating an acceleration and a slowing of conduction velocity during high-frequency firing for control and ZD7288, respectively. The difference in delay of the 20th action potential was maximal at intermediate frequencies (200 and 333 Hz; *Figure 4G*). These experiments show, that HCNs, despite their slow kinetics, ensure reliable high-frequency firing.

## The passive membrane properties of cMFBs are HCN- and cAMP-dependent

To better understand how $I_h$ impacts action potential firing, we next investigated the passive membrane properties of cMFBs by recording the voltage response elicited by small hyperpolarizing current injections (–10 pA for 300 ms) in the absence and presence of 30 µM ZD7288 (*Figure 5A and B*). ZD7288 caused (i) a hyperpolarization of the resting membrane potential by, on average, 5.4 mV ($-80.0 \pm 0.6$ mV and $-85.4 \pm 1.4$ mV for control and ZD7288-treated cMFBs n = 94 and 35, respectively), (ii) a doubling of the apparent input resistance calculated from the steady-state voltage at the end of the current step ($794 \pm 48$ MΩ and $1681 \pm 185$ MΩ, respectively), and (iii) a doubling of the apparent membrane time constant, as determined by a mono-exponential fit to the initial decay of the membrane potential ($14.4 \pm 0.8$ ms and $35.0 \pm 2.5$ ms, respectively; see legend of *Figure 5B* for statistical testing). To analyze the cAMP-dependence of the conduction velocity (cf. *Figure 1*),

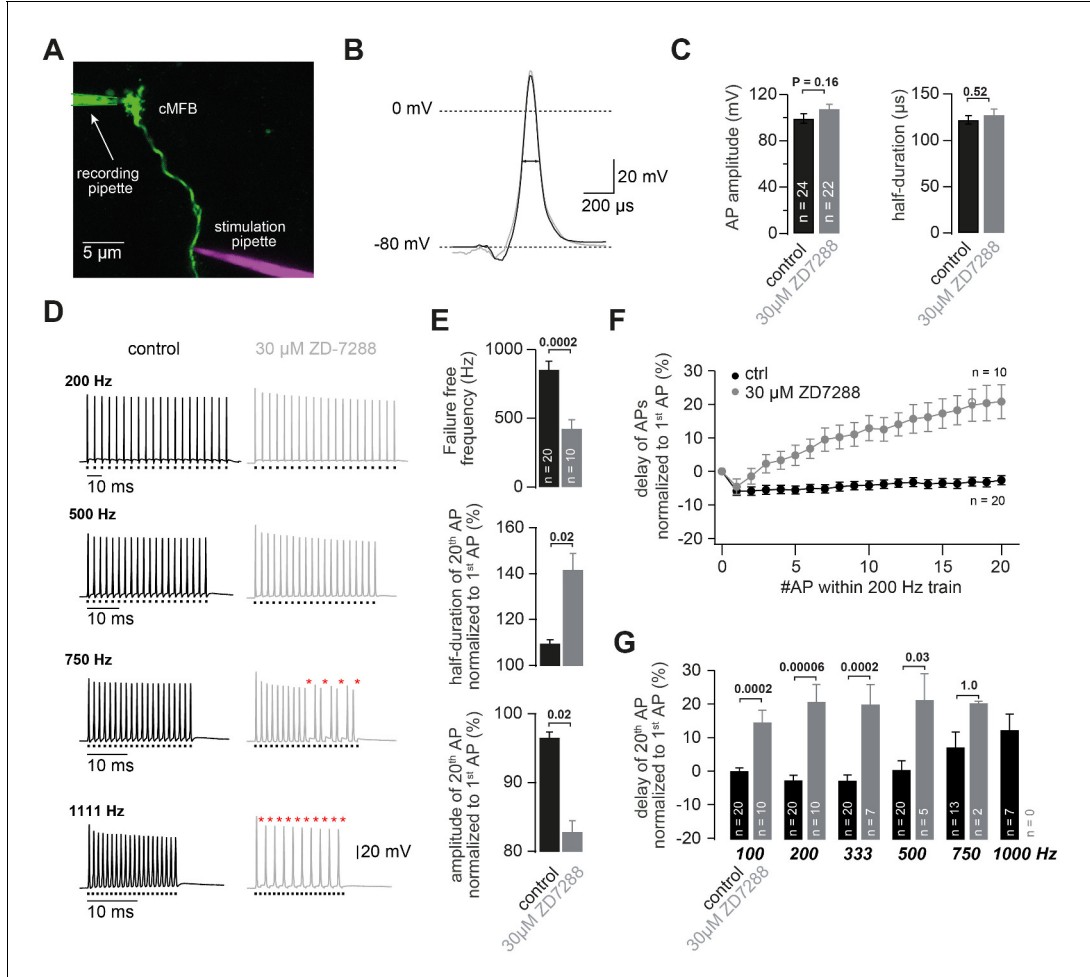

**Figure 4.** HCN channels support high-frequency action potential firing. (**A**) Two-photon microscopic image of a whole-cell patch-clamp recording from a cMFB (green) filled with the fluorescent dye Atto 488 in an acute cerebellar brain slice of an adult (43-day-old) mouse (maximal projection of stack of images). Targeted axonal stimulation was performed by adding a red dye, Atto 594, to the solution of the stimulation pipette. (**B**) Grand average of action potentials evoked at 1 Hz under control conditions (black) and in the presence of ZD7288 (gray). (**C**) Average action potential amplitude (measured from resting to peak) and half-duration ($P_{T-Test}$ = 0.16 and 0.51 for amplitude and resting, respectively). (**D**) Example traces of two different cMFBs stimulated at frequencies between 200 Hz and 1111 Hz under control conditions (*left, black*) or in the presence of 30 μM ZD7288 (*right, gray*). Traces for 100, 333, 1000 and 1666 Hz are not shown. The time of stimulation is indicated below each trace. Failures are illustrated by red asterisks. (**E**) (*Top graph*) Average maximal failure-free firing frequency for control (*black*) and ZD7288-treated (*gray*) cMFBs. (*Middle and bottom graph*) Average amplitude reduction and action potential broadening of the 20th compared with the 1st action potential (AP) of trains of 20 stimuli at 200 Hz for control (*black*) and ZD7288-treated(*gray*) cMFBs. (**F**) Average delay between the peak of the APs and the stimulation during trains of 20 stimuli at 200 Hz, normalized to the delay of the first AP, for control conditions (*black*) and ZD7288-treated (*gray*) cMFBs. (**G**) Average delay of the 20th AP normalized to the delay of the 1st action potential during failure-free trains of 20 stimuli at frequencies ranging from 100 to 1000 Hz. The P-values were obtained from t-tests and were multiplied by six to apply a Bonferroni-correction, indicating a highly significant slowing of the conduction velocity during failure-free high-frequency trains in ZD7288-treated cMFBs compared with controls. Note that the number of experiments decreased with increasing frequency because the analysis harestricted to failure-free traces.

DOI: https://doi.org/10.7554/eLife.42766.006

we determined the cAMP-dependence of the passive membrane properties of cMFBs. Adding cAMP in various concentrations to the intracellular solution depolarized the membrane potential and decreased both the input resistance and the apparent membrane time constant in a concentration-dependent manner. These effects are opposite to those that result from the application of ZD7288 (*Figure 5B*). These data suggest that HCN channels in cerebellar mossy fibers determine the passive membrane properties as a function of the intracellular cAMP concentration.

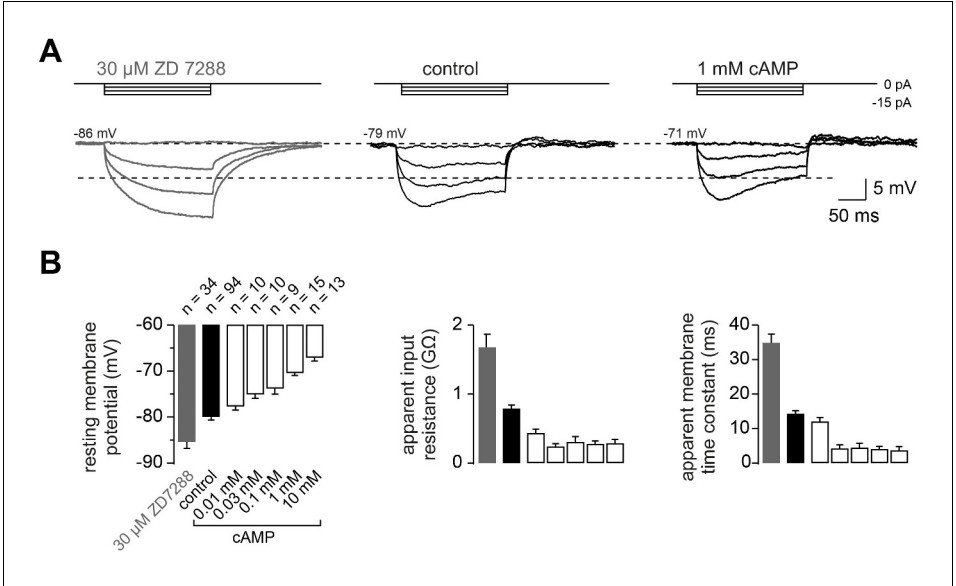

**Figure 5.** The passive membrane properties of cMFBs are HCN- and cAMP-dependent. (**A**) Example voltage response of cMFBs to small hyperpolarizing current steps. The application of 30 µM ZD7288 eliminated the $I_h$-mediated voltage sag (left). Adding 1 mM cAMP to the intracellular path-clamp solution (right) reduced the input resistance as seen by the reduced steady-state voltage response (dashed lines). (**B**) Average resting membrane potential (left), apparent input resistance (middle), and apparent membrane time constant (right) upon application of 30 µM ZD7288 or different concentrations of cAMP. For all three parameters, $P_{ANOVA}$ and $P_{Kruskal-Wallis}$ are $<10^{-10}$. The Dunnett test for multiple comparisons with the control indicates significance for control vs. ZD7288 (p<0.0001) and for control vs. cAMP concentrations $>\sim0.1$ mM (e.g., P<0.001 for control vs. 1 mM cAMP).
DOI: https://doi.org/10.7554/eLife.42766.007

## HCN2 is uniformly distributed in mossy fiber axons and boutons

Of the four HCN subunits (HCN1–HCN4), the subunits HCN1 and HCN2 are predominantly expressed in the cerebellar cortex (*Notomi and Shigemoto, 2004*; *Santoro et al., 2000*). Previous studies in the cortex, hippocampus, and auditory brainstem primarily detected HCN1 in axons (*Elgueta et al., 2015*; *Huang et al., 2011*; *Ko et al., 2016*), but HCN2 was found to be more sensitive to cAMP than HCN1 (*Wang et al., 2001*; *Zagotta et al., 2003*). To understand the pronounced cAMP-dependence of conduction velocity (cf. *Figure 1*) and the passive membrane properties (cf. *Figure 5*) at the molecular level, we investigated the identity and distribution of HCN channels using pre-embedding immunogold labeling for HCN1 and HCN2 in cMFBs and adjacent axons. At the electron microscopic level, we found only background immunoreactivity for HCN1 (data not shown) but significant labeling for HCN2 (*Figure 6A*). HCN2 immunogold particles were diffusely distributed along the plasma membrane of cMFBs, with similar labeling density in the adjacent mossy fiber axon (*Figure 6B*), which could be traced back up to 3.5 µm from cMFBs. In addition, we created a 3D reconstruction of a cMFB (*Figure 6C* and *Figure 6—video 1*), including gold particles for HCN2 and identified synaptic connections. Synapses onto granule cell dendrites were observed within invaginated parts of the bouton. HCN2 was uniformly distributed without apparent spatial relations to those synapses. The density of immunogold particles for HCN2 in this reconstructed bouton was 17.1 particles/µm$^2$ (in total 1260 particles per 73.65 µm$^2$). The mean density of immunogold particles for HCN2 was 22.7 ± 2.4 per µm$^2$ (n = 6 cMFBs from two mice). These data indicate that HCN2 is the dominant subunit mediating $I_h$ in cMFBs, consistent with its pronounced cAMP-dependence.

## HCN channels in cMFB are strongly modulated by cAMP

To better understand the function of axonal HCN2 channels and their modulation by intracellular cAMP, we performed voltage-clamp recordings from cMFBs with different cAMP concentrations in the intracellular patch solution. Hyperpolarizing voltage steps evoked a slowly activating, non-inactivating inward current, which was inhibited by ZD7288 (*Figure 7A*). Using the tail currents of

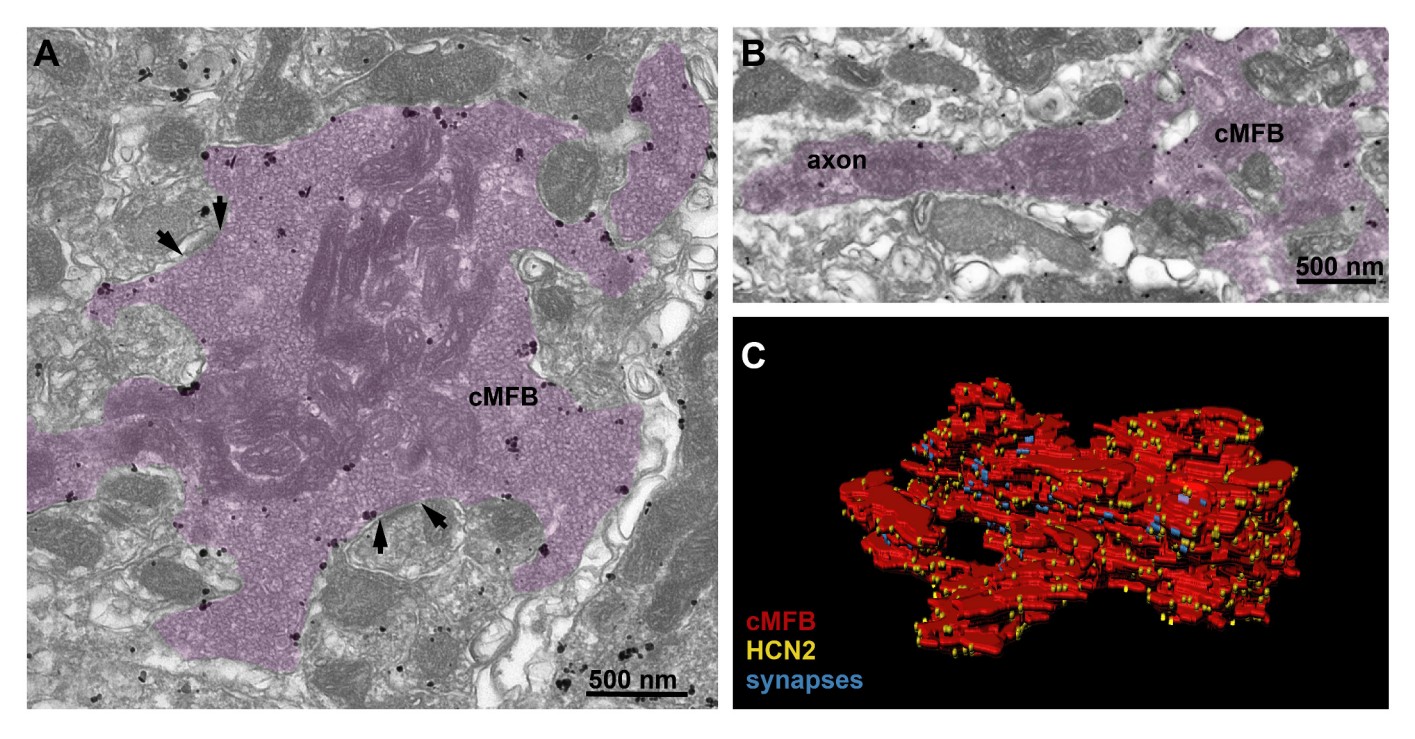

**Figure 6.** HCN2 is uniformly distributed in mossy fiber axons and boutons. (**A**) Electron microscopic image showing a cMFB (magenta) labeled for HCN2. Many particles are diffusely distributed along the plasma membrane of the cMFB, some of them being clustered. Arrows mark synapses between the cMFB and dendrites of adjacent granule cells. (**B**) Another cMFB, showing similar labeling density for HCN2 in a proximal part of the mossy fiber axon. (**C**) Reconstructed cMFB (red) with identified synapses based on the ultrastructure (blue) and with HCN2 labeled with gold particles (yellow).

DOI: https://doi.org/10.7554/eLife.42766.008
The following video is available for figure 6:
**Figure 6—video 1.** Reconstructed cMFB with labeled synapses and HCN2 channels.
DOI: https://doi.org/10.7554/eLife.42766.009

ZD7288-sensitive currents evoked by voltage steps between –80 mV and –150 mV from a holding potential of –70 mV, we calculated the activation curve of $I_h$ with a mean $V_{1/2}$ of –103.3 ± 0.8 mV (**Figure 7B**; n = 36 $V_{1/2}$-values, each from a different cMFB). On the basis of the average resting membrane potential of cMFBs, this means that about 7% of the overall HCN2-mediated current is active at rest.

To analyze the cAMP concentration-dependence of $I_h$, we added different concentrations of cAMP (30 μM to 10 mM) to the intracellular patch solution. With 1 mM cAMP, $V_{1/2}$ shifted by 17 mV to, on average, –86.6 ± 1.2 mV (n = 16 cMFBs; $P_{T-Test} < 10^{-10}$; **Figure 7B**). The resulting average shifts of $V_{1/2}$ revealed an $EC_{50}$ of 40.4 μM intracellular cAMP (**Figure 7D**). In order to estimate the endogenous presynaptic cAMP concentration, we performed presynaptic perforated-patch recordings on cMFBs. Under perforated patch conditions, the $V_{1/2}$ of $I_h$ was –96.4 ± 1.2 mV (n = 10), significantly more depolarized than the corresponding whole-cell recordings after rupture of the perforated patch (–101.3 ± 1.0 mV; n = 10; $P_{T-Test}$ = 0.0076; **Figure 7C**; see Materials and methods for comparisons with additional control groups). This voltage shift (4.9 ± 1.2 mV, n = 10 cMFBs) indicates an endogenous cAMP concentration of 12.6 μM in cMFBs, with a 68% confidence interval of 1.8 to 60.7 μM cAMP (**Figure 7D**). These data reveal a high endogenous resting cAMP concentration.

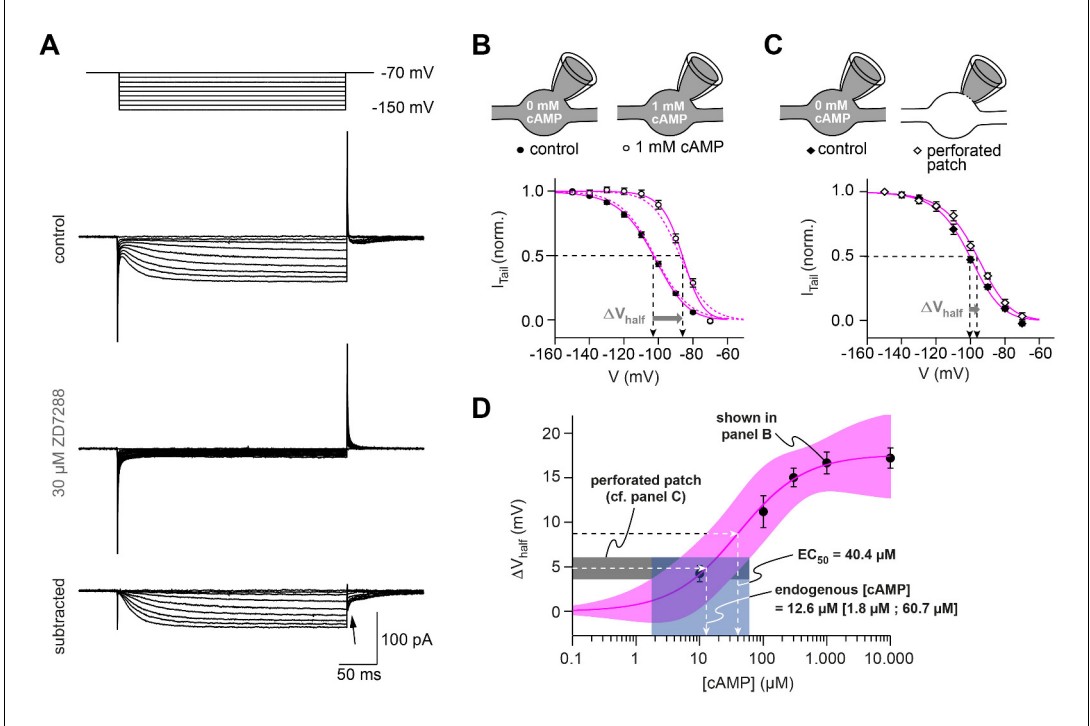

**Figure 7.** HCN channels in cMFB are strongly modulated by cAMP. (A) Example currents elicited by hyperpolarizing voltage steps (from –70 mV to a voltage between –70 mV and –150 mV). *Top*, control current, *middle,* remaining transients in the presence of 30 μM ZD7288 and, *bottom*, subtracted currents. The ZD7288-sensitive current is slowly activating, non-inactivating and shows inward tail currents (arrow). (B) Activation curve of $I_h$ determined as the normalized tail current of ZD7288-sensitive currents obtained after the end of the conditioning voltage pulse (arrow in panel (A)) plotted against the corresponding voltage pulse with 0 mM cAMP (filled circles, n = 36) and 1 mM cAMP (open circles, n = 15) in the intracellular solution. Sigmoidal fits (continuous magenta lines) yield the midpoints of $I_h$ activation ($V_{1/2}$, arrows). The steady-state activation curves produced by the Hodgkin-Huxley models (dotted magenta line) are superimposed. *Inset on top*: illustration of the whole-cell recording configuration with 0 and 1 mM cAMP in the intracellular solution. (C) Activation curves obtained with the perforated-patch recordings and after rupture of the perforated membrane patch (n = 10). *Inset on top*: illustration of the whole-cell recording configuration with 0 mM cAMP in the intracellular solution and in the perforated patch configuration, when the intracellular cAMP concentration is unperturbed. (D) Shift in $I_h$ $V_{1/2}$ versus the corresponding cAMP concentration (mean ± SEM). Fitting the data with a Hill equation (magenta line) revealed an $EC_{50}$ of 40.4 μM. Superposition of the 68% confidence band of the fit (light magenta area) with the average voltage shift observed in perforated patch recordings (4.8 ± 1.2 mV, n = 10, dotted black line and gray area) results in an estimated endogenous cAMP-concentration of 12.6 μM with a 68% confidence interval of 1.8 to 60.7 μM cAMP (dotted line and light blue area).
DOI: https://doi.org/10.7554/eLife.42766.010

## Hodgkin-Huxley model describing HCN2 channel gating

For our ultimate aim, to obtain a mechanistic and quantitative understanding of axonal HCN2 function in cerebellar mossy-fiber axons, we developed a computational Hodgkin-Huxley (HH) model. The model was constrained to the experimentally recorded $I_h$ kinetics derived from the activation and deactivation time constants of $I_h$ (*Figure 8A*) measured at potentials between –70 and –150 mV. The activation curve (cf. *Figure 8B*), as well as the averaged time constants for both activation (n = 20) and deactivation (n = 15; *Figure 8B*), were well described by a HH-model with one activation gate. In addition, we generated an alternative HH-model to describe the HCN2 current in the presence of 1 mM intracellular cAMP (for a more detailed implementation of the cAMP-dependence of HCN2 gating, see *Hummert et al., 2018*). Furthermore, we estimated the reversal potential of $I_h$ with short voltage ramps as described previously (*Cuttle et al., 2001*) and found a value of –23.4 ± 1.4 mV (n = 7; *Figure 8C*), similar to previous estimates (*Aponte et al., 2006*; *Cuttle et al., 2001*). These data provide a quantitative description of axonal $I_h$ at cMFBs.

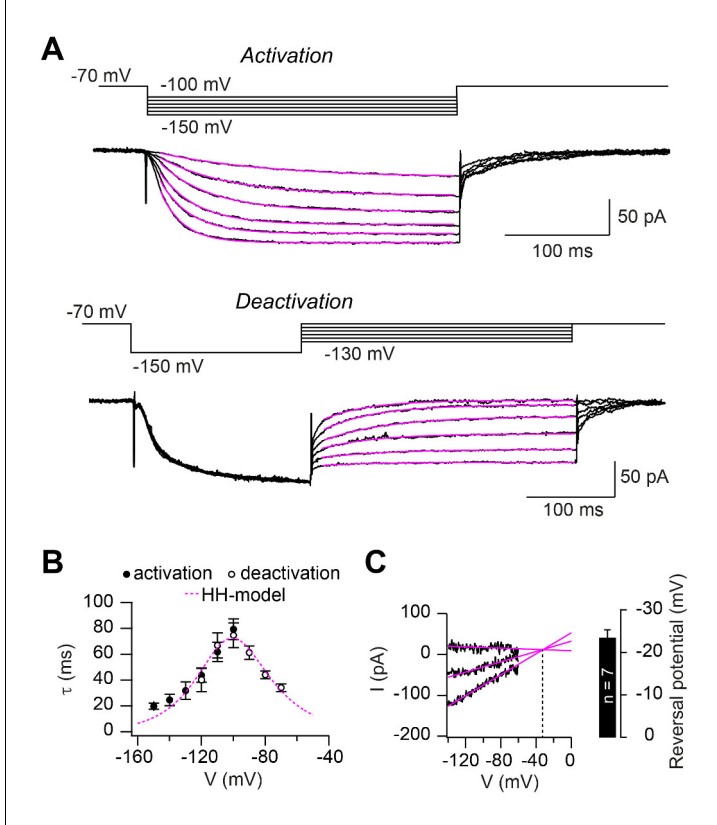

**Figure 8.** Hodgkin-Huxley model describing HCN2 channel gating. (**A**) Example of ZD7288-sensitive currents (black) elicited by the illustrated activation (*top*) and deactivation (*bottom*) voltage protocols superimposed with mono-exponential fits (magenta). (**B**) Average time constants of activation (filled circles) and deactivation (open circles; mean ± SEM). The dotted magenta line represents the prediction of $I_h$ activation and deactivation time constant based on the Hodgkin-Huxley model. (**C**) Example of linear extrapolation (magenta lines) of leak subtracted currents evoked by fast (10 ms) voltage ramps generated from a range of holding potentials that extended across the activation range of $I_h$. The reversal potential was found to be –36 mV in this example. *Inset:* average reversal potential from seven independent experiments.
DOI: https://doi.org/10.7554/eLife.42766.011

## Mechanism of control of conduction velocity and metabolic costs of HCN channels

What are the mechanisms by which axonal HCN2 channels control conduction velocity? In principle, the depolarization caused by HCN2 channels will bring the resting membrane potential closer to the threshold for voltage-gated Na$^+$ channel activation, which could accelerate the initiation of the action potential (see discussion). Alternatively, the increased membrane conductance caused by HCN2 channels will decrease the membrane time constant, which could accelerate the voltage responses, as has been shown, for example, for dendritic signals in auditory pathways (*Golding and Oertel, 2012*; *Mathews et al., 2010*). To distinguish between these two possibilities, we generated a conductance-based NEURON model consisting of cylindrical compartments representing cMFBs that are connected by myelinated axons (*Figure 9A*; *Ritzau-Jost et al., 2014*). The model contained voltage-dependent axonal Na$^+$ and K$^+$ channels, passive Na$^+$ and K$^+$ leak channels, and the established HH model of $I_h$ (cf. *Figure 8*). After adjustments of the peak conductance densities, the model captured the current clamp responses to –10 pA current injections (*Figure 9B*), the resting membrane potential (*Figure 9C*), and the apparent input resistance (*Figure 9D*). Removing the HH model of $I_h$, or replacing it with the 1-mM-cAMP-HH-model of $I_h$, reproduced the corresponding voltage responses, the shift in the resting membrane potential, and the change in the apparent input resistance obtained in the presence of ZD7288 or 1 mM intracellular cAMP (*Figure 9B–D*). Interestingly,

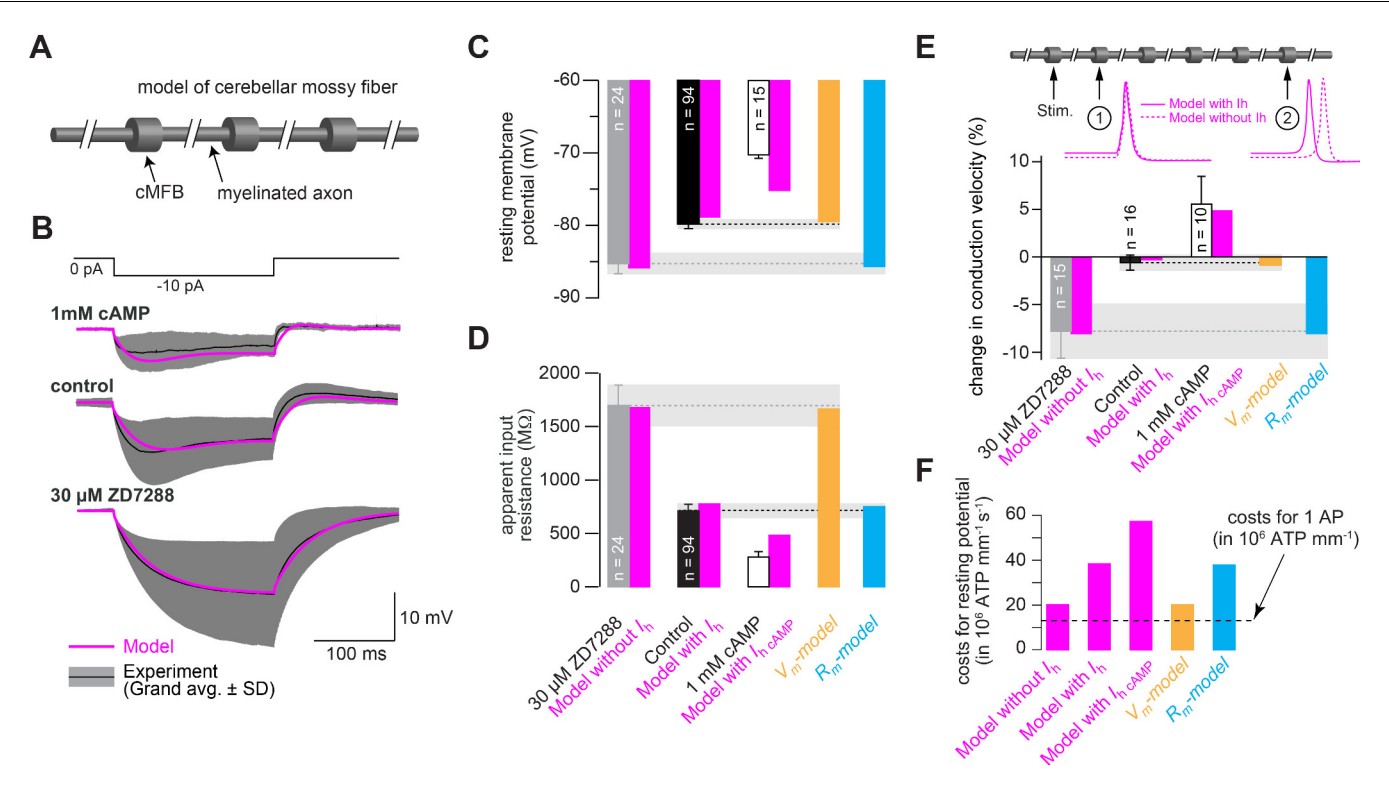

**Figure 9.** Mechanism of control of conduction velocity and metabolic costs of HCN channels. (A) Illustration of the cerebellar mossy fiber model consisting of 15 connected cylindrical compartments representing cMFBs and the myelinated axon. (B) Grand average voltage response (black) and standard deviation (gray area) of cMFBs to a –10 pA hyperpolarizing current pulse with 1 mM cAMP included in the patch pipette (top), under control conditions (middle) or for cMFBs treated with ZD7288 (bottom), superimposed with the predicted voltage response from the model (magenta). (C) Average resting membrane potential of cMFBs measured under control conditions (black), with ZD7288 (gray), or 1 mM intracellular cAMP (open bar; data from *Figure 5B*) compared to the predictions from the corresponding models (magenta). Furthermore, the chart also shows the resting membrane potential of the two models that simulate only the membrane depolarization ($V_m$- model; light brown) or only the decreased membrane resistance ($R_m$-model; blue) caused by HCN channels. (D) Corresponding comparison between the measured values and the predictions from the models as shown in panel (C) for the apparent input resistance of cMFBs. (E) Corresponding comparison between the measured values and the predictions from the models as shown in panels (C) and (D) for the conduction velocity in mossy fibers. *Inset top:* illustration of the model of a mossy fiber and the action potentials at two different positions with (magenta line) and without (dashed magenta line) the HH model of HCN channels. (F) The calculated metabolic costs for maintaining the resting membrane potential are shown for each model as the number of required ATP molecules per mm of mossy fiber axon and per second. The metabolic cost of the firing of a single action potential (AP) is indicated by the dashed line as the number of required ATP molecules per mm of mossy fiber axon (this number was very similar for all models).

DOI: https://doi.org/10.7554/eLife.42766.012

The following figure supplement is available for figure 9:

**Figure supplement 1.** Impact of depolarization on Na$_V$ availability and on conduction velocity in our model of a mossy fiber axon.

DOI: https://doi.org/10.7554/eLife.42766.013

the models predicted a decrease of the conduction velocity when the control HH model was removed, and conversely an increase with the 1-mM-cAMP-HH model (*Figure 9E*), that was similar in extent to that measured experimentally with ZD7288 and 8-Br-cAMP treatments (cf. *Figure 1*). These findings support our conclusion that HCN2 channel modulation suffices to tune conduction velocity bidirectionally.

Next, we generated two additional models, in which either only the depolarizing effect of HCN2 channels ($V_m$-model) or the decreased input resistance (i.e. the decreased membrane resistance, $R_m$-model) was implemented (by modifying the K$^+$ or the $I_h$ reversal potential, respectively; see Materials and methods). The results showed that the $V_m$-model but not the $R_m$-model caused an increase in conduction velocity, indicating that the depolarizing effect of axonal HCN2 channels determines conduction velocity (*Figure 9E*). Interestingly, increasing the resting membrane

potential from –90 mV to –65 mV, decreased the availability of voltage-dependent $Na^+$ ($Na_V$) channels but increased the conduction velocity (*Figure 9—figure supplement 1*). The conduction velocity decreased only at resting membrane potentials above –65 mV in our model. Together, these data indicate that the depolarization mediated by HCN2 channels accelerates the conduction velocity by bringing the membrane potential closer to the firing threshold.

The non-inactivating nature of HCN channels and the accompanying shunt at the resting membrane potentials suggest that $I_h$ is metabolically expensive. Therefore, we calculated the $Na^+$ influx in each model and converted it into the ATP consumption required to restore the $Na^+$ gradient (*Hallermann et al., 2012*). Computational modeling showed that it is ~100% more expensive to maintain the resting membrane potential with $I_h$ than without or by depolarization alone ($V_m$-model; *Figure 9F*). Furthermore, the metabolic cost of maintaining the resting membrane potential with $I_h$ for one second was ~3-fold higher than the cost of generating one action potential (*Figure 9F*). Assuming an average frequency of cerebellar mossy fibers of 4 Hz in vivo (*Chadderton et al., 2004*; *Rancz et al., 2007*), the HCN2 channels increased the required energy of cerebellar mossy fibers by ~30%. With increasing firing frequency, the metabolic costs of action potential firing will become dominant compared with the HCN2-mediated costs for resting membrane potentials (e.g., ~3% at 40 Hz). These data indicate that HCN2 channels are a major consumer of the energetic demands of axons.

## Discussion

Here, we demonstrate that the HCN channels increase action potential velocity and fidelity in central axons. By combining electrophysiological, electron-microscopic, and computational techniques, we reveal the mechanism and the metabolic costs of the dynamic control of the velocity and fidelity of action potential propagation by HCN channels in the vertebrate central nervous system.

### Dynamic control of conduction velocity

We describe both an increase and decrease of the baseline axonal conduction velocity in the range of ~5% mediated by HCN channels (*Figures 1–3*). Furthermore, HCN channels increase the maximal failure-free firing frequency by a factor of two (*Figure 4*). Although the changes in baseline conduction velocity are relatively small, considering the long distances that axons traverse in the brain, HCN channels can be expected to change the arrival time of the action potential by, for example, 0.5 ms in the case of unmyelinated cerebellar parallel fibers (assuming 3 mm length and 0.3 m/s velocity; *Swadlow and Waxman, 2012*). Such temporal delays will influence information processing in the central nervous system, because spike-timing dependent plasticity (*Caporale and Dan, 2008*), coincidence detection (*Softky, 1994*), and the neuronal rhythms of cell ensembles (*Buzsáki et al., 2013*) precisely tune the arrival times of action potentials. There are several examples of the specific tuning of conduction velocity in the sub-millisecond domain: the diameter and the degree of myelination of cerebellar climbing fibers (*Sugihara et al., 1993*; *Lang and Rosenbluth, 2003*; but see *Baker and Edgley, 2006*), the degree of myelination of thalamocortical axons (*Salami et al., 2003*), and the internode distance of auditory axons (*Ford et al., 2015*) are all tuned exactly to offset the different arrival times of action potentials with a temporal precision of ~100 µs.

The cerebellum is involved in the accurate control of muscle contraction with a temporal precision of 1–100 ms (*Hore et al., 1991*). Submillisecond correlations in spike timing occurring between neighboring Purkinje cells have been noted previously (reviewed in *Isope et al., 2002*; *Person and Raman, 2012*). Furthermore, submillisecond precision of the mossy or parallel fiber input are critical for information processing in the cerebellar circuits (*Braitenberg et al., 1997*; *Heck et al., 2001*; *Isope et al., 2002*). Together, the here-described changes in action potential conduction velocity in mossy and parallel fibers (*Figures 1–3*) may thus play an important role in cerebellar computation. Furthermore, HCN channels facilitate high-frequency firing (*Figure 4*), which occurs in many parts of the mammalian CNS (*Delvendahl and Hallermann, 2016*).

Our findings that the cAMP-HCN pathway and neuromodulators can finely tune conduction velocity in the vertebrate central nervous system adds to the emerging idea that axons directly contribute to computation in neuronal circuits. Indeed, the view of the axon as a cable-like compartment in which conduction velocity is static has substantially changed over recent years in favor of a model that allows flexibility and complex forms of axonal computation (*Debanne et al., 2011*). Recent

findings showed that axon diameters change during long-term potentiation (*Chéreau et al., 2017*), and changes in myelination in the motor cortex were resolved during the learning of complex motor skills (*McKenzie et al., 2014*; for environmental effects on myelination see also *Forbes and Gallo, 2017*). One caveat of our study is the rather high concentration of the used neuromodulators and the lack of in vivo evidence for the neuromodulation of conduction velocity. However, our data demonstrate that, under certain conditions, an active control of conduction velocity could occur in the vertebrate CNS via the cAMP-HCN pathway.

## Mechanism and metabolic costs of HCN-channel-mediated control of action potential propagation

Our analysis revealed that the control of conduction velocity is solely mediated by changes in resting membrane potential. Isolated changes in membrane conductance and thus in the membrane time and length constant had no effect on conduction velocity (*Figure 9E*). The increased conduction velocity upon depolarization is consistent with a previously observed correlation between conduction velocity and the depolarization from the resting potential required to reach the firing threshold in motoneurons (*Carp et al., 2003*). On the other hand, $Na^+$ channels have a steep steady-state inactivation and are partially inactivated at the resting membrane potential in axons (*Battefeld et al., 2014*; *Engel and Jonas, 2005*; *Rama et al., 2015*). Depolarization could thus be expected to further inactivate $Na^+$ channels and to decrease conduction velocity. However, our modeling results showed that increasing the membrane potential from –90 to –60 mV increased the conduction velocity despite significantly decreasing $Na^+$ channel availability (*Figure 9—figure supplement 1*). Interestingly, these findings are in agreement with the nonlinear cable theory predicting that the difference between the resting membrane potential and the firing threshold is a critical parameter for action potential conduction velocity (see, for example, Figure 12.25 in *Jack et al. (1983)* which shows increasing velocity with increasing safety factor, that is, decreasing excitation threshold '$V_B$'). Intuitively, the HCN-channel-mediated acceleration of conduction velocity can be understood as follows: $Na_V$-mediated current influx in one axonal location will depolarize neighboring locations faster above the threshold in a depolarized axon compared with a resting axon. In our model, this effect outweighs the disadvantage of the increased steady-state inactivation of $Na^+$ channels up to a membrane potential of about –65 mV and a $Na_V$ availability of 50%. But the exact values above which $Na_V$ availability limits conduction velocity critically depend on the assumptions of the model, such as the voltage-dependence of inactivation and the density of the $Na_V$ channel. Interestingly, $Ca^{2+}$ entering through axonal voltage-gated $Ca^{2+}$ channels (*Brenowitz and Regehr, 2007*) could interact with the cAMP pathway by activating or inhibiting different subtypes of adenylyl cyclase and phosphodiesterase (*Bruce et al., 2003*).

We observed a marked decrease in the maximal failure-free firing frequency from ~800 to ~400 Hz in cerebellar mossy fiber axons upon blockade of HCN channels (*Figure 4*). Although we cannot differentiate between initiation failure and conduction failure, the alterations in the half duration and amplitude of action potentials during high-frequency firing (*Figure 4E*) argue for impaired action potential conduction in the absence of HCN channels. This is consistent with findings in cerebellar parallel fibers and hippocampal Schaffer collaterals, where HCN channels ensure reliable conduction, particularly at branch points (*Baginskas et al., 2009*; *Soleng et al., 2003*). The extent to which conduction failures occur under physiological conditions is controversial (*Debanne et al., 2011*; *Rama et al., 2018*; *Radivojevic et al., 2017*), but our data indicate that HCN channels are required to ensure the reliable initiation and conduction of action potentials at high frequencies. This is consistent with $I_h$ counteracting the hyperpolarization during high-frequency firing (*Waxman et al., 1995*).

Our modeling results (*Figure 9*) indicate that the evolutionary design of HCN channels as a continuously open shunt for $Na^+$ influx incurs significant metabolic costs. These high costs might appear surprising, because a metabolically cheaper way to depolarize the membrane would be the expression of fewer $Na^+$-$K^+$-ATPases, resulting in a depolarized $K^+$ reversal potential (cf. $V_m$-model in *Figure 9*). However, such a design might complicate high-frequency firing. Furthermore, as discussed in the following paragraph, our finding that conduction velocity can be rapidly regulated via the cAMP-HCN pathway might provide an additional justification for the metabolic costs of axonal HCN channels.

## Modulation of conduction velocity via intracellular cAMP concentration

Using direct whole-cell recordings and immunogold EM from *en passant* boutons in cerebellar axons, we identified near exclusive expression of HCN2 isoforms and a half-maximal shift of the activation of HCN2 channels at a cAMP concentration of 40 µM (*Figure 7D*). Furthermore, our perforated patch-recordings from axonal compartments provide, to our knowledge, the first direct estimate of endogenous cAMP concentration in vertebrate central axons of 13 µM (*Figure 7D*). This is higher than previous estimates of 50 nM in Aplysia sensory neurons (*Bacskai et al., 1993*; but see *Greenberg et al., 1987*) and 1 µM in cardiomyocytes (*Börner et al., 2011*). But a recently reported low cAMP-sensitivity of protein kinase A (*Koschinski and Zaccolo, 2017*), a prototypical cAMP-regulated protein, also argues for high intracellular cAMP concentrations. On the other hand, our data do not rule out that such high cAMP concentrations are limited to spatially restricted domains. The possibility of local cAMP signaling-compartments was recently observed in *Drosophila* axons (*Maiellaro et al., 2016*).

A high endogenous cAMP concentration and expression of the HCN2 isoform facilitates the ability of neuromodulators to control conduction velocity bidirectionally and dynamically. Only norepinephrine increased the conduction velocity in cerebellar parallel fibers whereas the other neuromodulators reduced the velocity (*Figure 2*), consistent with the expression of both $G_i$- and $G_s$-coupled receptors, respectively. Indeed, $G_s$-coupled receptors for serotonin, dopamine, and adenosine are expressed in the molecular layer of the cerebellum (see, for example, *Geurts et al., 2002*; *Schweighofer et al., 2004*). Interestingly, adenosine, which decreased the conduction velocity (*Figure 2*), has been shown to be an endogenous sleep factor (*Basheer et al., 2004*; *Porkka-Heiskanen et al., 1997*). Moreover, serotonin, dopamine, and norepinephrine play important regulatory functions during sleep in, for example, the cerebellum (*Canto et al., 2017*). Therefore, it is tempting to speculate that the cAMP-HCN pathway allows not only an increase in the conduction velocity during arousal but also a decrease in the velocity that saves metabolic costs during periods of rest or sleep. The cAMP-HCN pathway in axons could thus contribute to the reduced energy consumption of the brain during sleep (*Boyle et al., 1994*; *Townsend et al., 1973*). It should be noted that the observed modulation of conduction velocity by neurotransmitters (*Figure 2*) is consistent with a modulation via the cAMP-HCN pathway. Nevertheless, other mechanisms, such as direct influences on voltage-dependent $Na^+$ (*Yin et al., 2017*), $K^+$ (*Yang et al., 2013*), and $Ca^{2+}$ channels (*Burke et al., 2018*), could contribute to the modulation of conduction velocity. Furthermore, off-target interactions cannot be excluded with the used concentrations of neuromodulators.

## Clinical relevance of axonal HCN channels

The function of HCN channels has been studied in human peripheral nerves using non-invasive threshold tracking techniques (*Howells et al., 2016*; *Howells et al., 2012*; *Lorenz and Jones, 2014*). Significant alterations of HCN channel expression and/or function have been described in pathologies such as stroke (*Jankelowitz et al., 2007*), porphyria (*Lin et al., 2008*), diabetic neuropathy (*Horn et al., 1996*), neuropathic pain (*Chaplan et al., 2003*), and inflammation (*Momin and McNaughton, 2009*), as well as in a vertebrate model of demyelination (*Fledrich et al., 2014*). In some of these cases, the alterations are consistent with an activity-dependent modulation of HCN channels (*Jankelowitz et al., 2007*). Furthermore, HCN channels seem to be causally related to pain symptoms (*Chaplan et al., 2003*; *Momin and McNaughton, 2009*) and therapeutic blockade of HCN channels are also considered (*Wickenden et al., 2009*). On the basis of our findings, HCN could also play a compensatory role in restoring conduction velocity in some diseases.

# Materials and methods

## Preparation of cerebellar slices

Cerebellar slices were prepared from P21-P46 C57BL/6 mice of either sex as reported previously (*Ritzau-Jost et al., 2014*). In short, after anesthetization with isoflurane, mice were killed by rapid decapitation; the cerebellar vermis was quickly removed and placed in a slicing chamber filled with ice-cold extracellular solution (ACSF) containing (in mM): NaCl 125, KCl 2.5, NaHCO₃ 26, NaH₂PO₄ 1.25, glucose 20, CaCl₂ 2, MgCl₂ 1 (pH adjusted to 7.3–7.4 with HCl). Parasagittal or horizontal slices were cut from the vermis of the cerebellum using a microtome with a vibrating blade (VT1200, Leica

Biosystems, Nussloch, Germany), incubated at 35°C for approximately 30 min and subsequently stored at room temperature until use. For electrophysiological recordings, a slice was transferred into the recording chamber mounted on the stage of an upright Nikon microscope. The recording chamber was perfused with ACSF and the temperature in the center of the recording chamber was set to 35°C using a TC-324B perfusion heat controller (Warner Instruments, Hamden CT, USA).

## Measuring conduction velocity in cerebellar parallel and mossy fibers

Compound action potentials were evoked by electrical stimulation using a bipolar platinum/iridium electrode (from Microprobes for Life Science, Gaithersburg MD, USA) placed either in the white matter or in the molecular layer (*Figure 1*) of the cerebellum. For the extracellular recording of compound action potentials, two pipettes were filled with a 1M NaCl solution (tip resistance of 1–3 MΩ) and placed within the respective fiber bundle and the voltage was measured in current clamp mode with an EPC10 amplifier (CC gain 10x). Compound action potentials were evoked at 0.5 and 1 Hz in parallel and mossy fibers, respectively. All recordings were performed in the presence of 10 μM NBQX to block synaptic potentials. The conduction velocity of parallel fibers was measured at 35°C. Owing to the higher conduction velocity in myelinated mossy fibers, the action potentials evoked by white matter stimulation had to be recorded at room temperature to allow the separation of the compound action potential from the stimulation artifact. To calculate the conduction velocity, we determined the delays of the peaks of the compound action potential component recorded with the proximal and distal electrode. Compound action potentials from mossy fibers were analyzed offline using the smoothing spline interpolation operation of Igor Pro to increase the signal to noise ratio. Control recordings were performed interleaved with the application of different drugs. The conduction velocity experienced a small rundown over 20 min under control conditions (*Figures 1* and *2C–E*). To investigate the contribution of HCN channels on the neuromodulation of the conduction velocity (*Figure 2F–G*), cerebellar slices were pre-incubated for 20 min at room temperature with 30 μM ZD7288 and then transferred to the recording chamber (35°C) with continued application of 30 μM ZD7288. The neuromodulators were applied 5 min after the beginning of the recordings (t = 0 min). Note that the control data without application of neuromodulators ('only ZD7288' in *Figure 2F and G*) show a larger decrease in conduction velocity during the 20 min recording period compared to the control data in *Figure 2C–E*. The difference in these control data is most likely due to the slow action of ZD7288 (cf. *Figure 1*), which further decreases the conduction velocity during the recordings, in addition to the run-down observed in control recordings without ZD7288. Therefore, the change in conduction velocity induced by the neuromodulators with and without pre-incubation of ZD7288 (*Figure 2F–G and C–E*, respectively) cannot be compared directly but have to be compared with the corresponding controls.

## Measuring conduction velocity in the optic nerve

Male wildtype mice of the C57BL6/N strain (P63±4) were euthanized by decapitation. After the brain was exposed, the optic nerves (ON) were separated from the retina at the ocular cavity, and both ONs were detached by cutting posterior to the optic chiasm. The preparation was gently placed into an interface brain/tissue slice (BTS) perfusion chamber (Harvard Apparatus) and continuously superfused with ACSF, bubbled with carbogen (95% $O_2$, 5% $CO_2$) at 36.5°C during the experiment (*Trevisiol et al., 2017*). In case both nerves were used for experiments, the non-recorded ON was transferred to a different incubation chamber (Leica HI 1210) that provided incubation conditions similar to those experienced by the recorded nerve while preventing exposure to ZD7288 and 8-Br-cAMP. The temperature was maintained constant using a feedback-driven temperature controller (model TC-10, NPI electronic) connected to a temperature probe (TS-100-S; NPI electronic) inserted into the BTS incubation chamber near the nerve. Each ON was detached from the optic chiasm and individually placed into the suction electrodes for stimulation and recording. The stimulation's direction of the ON was maintained constant (orthodromic) throughout the experiments by inserting the proximal (retinal) end of the nerve into the stimulation electrode as illustrated in *Figure 1I*. The stimulating electrode was connected to a battery (Stimulus Isolator A385; WPI) that delivered a supra-maximal stimulus to the nerve. The voltage was pre-amplified 500 times and fed to the AD ports of the EPC9 or acquired directly via the EPC9 headstage (HEKA Elektronik, Lambrecht/Pfalz). The reference channel was obtained from an ACSF-filled glass capillary next to the recording suction

electrode, which was in contact with the bathing ACSF. Initial equilibration of the ONs was performed at 0.1 Hz stimulation, until the recorded compound action potentials showed a steady shape (typically around 45–60 min from preparation). Five nerves from four animals and four nerves from four animals were used for ZD7288 and 8-Br-cAMP treatment, respectively. Compound action potentials were analyzed as described above using the smoothing spline interpolation operation of Igor Pro to increase the signal to noise ratio.

## Patch-clamp recordings from cMFBs

Recordings from cMFBs were visualized as previously described (*Ritzau-Jost et al., 2014*) with infrared differential interference contrast (DIC) optics using a FN-1 microscope from Nikon with a 100x objective (NA 1.1) or infrared oblique illumination optics using a Femto-2D two-photon microscope (Femtonics, Budapest) with a 60x Olympus (NA 1.0) objective. The passive properties of the cMFB were determined as previously described (*Hallermann et al., 2003*) and revealed similar values for a two-compartment model (data not shown) as previously described for cMFBs (*Ritzau-Jost et al., 2014*), indicating that we did indeed record from cMFBs. Furthermore, the access resistance was on average 16.9 ± 0.9 MΩ (n = 53 cMFBs), indicating optimal voltage clamp conditions.

To elicit traveling action potentials by axonal stimulation with a second pipette (*Figure 4A*), whole-cell recordings from cMFBs were performed with 50 µM of the green-fluorescent dye Atto488 (from Atto-Tec, Siegen, Germany) in the intracellular solution to visualize single mossy fiber axons. The additional stimulation pipettes filled with ACSF and 50 µM of the red-fluorescent dye Atto594 had the same opening diameter as patch pipettes and were positioned close to the axon and approximately 100 µm away from the patched terminal. Stimulation pulses with durations of 100 µs were delivered by a voltage-stimulator (ISO-Pulser ISOP1, AD-Elektronik, Buchenbach, Germany). The stimulation intensity (1–30 V) was adjusted to ensure failure-free initiation of action potentials at 1 Hz (~1.5 time the firing threshold). High-frequency trains of action potentials were evoked at 100, 200, 333, 500, 750, 1000, 1111 and 1666 Hz. Amplitudes were measured from peak to baseline. The duration was determined at half-maximal amplitude and is referred to as half-width. Action potentials were treated as failures if the peak did not exceed –40 mV.

Recordings were performed with an EPC10/2 patch-clamp amplifier, operated by the corresponding software PatchMaster (HEKA Elektronik) running on a personal computer. Recording electrodes were pulled from borosilicate glass capillaries (inner diameter 1.16 mm, outer diameter 2 mm) by a microelectrode puller (DMZ-Universal Puller, Zeitz Instruments, Augsburg). Pipettes used for patch-clamp recordings had open-tip resistances of 5–12 MΩ. The intracellular presynaptic patch pipette contained (in mM): K-gluconate 150, MgATP 3, NaGTP 0.3, NaCl 10, HEPES 10 and EGTA 0.05. The apparent input resistance of cMFBs was estimated by linear regression of the steady-state voltage in response to 300 ms hyperpolarizing current pulses of increasing amplitude (–five to –20 pA), whereas the apparent membrane time constant was determined by fitting the voltage response to a −10 pA hyperpolarizing pulse with a mono-exponential function.

$I_h$ activation curves determined from the analysis of normalized tail current were fitted with a Boltzmann function:

$$\frac{I}{I_{max}} = \frac{1}{1 + e^{\frac{V - V_{1/2}}{k}}},$$

where V is the holding potential, $V_{1/2}$ is the voltage of half-maximal activation and $k$ the slope factor. The reversal potential of $I_h$ was calculated from leak-subtracted currents evoked by 10 ms long voltage ramps extending across the activation range of $I_h$ (*Cuttle et al., 2001*). Three I-V relationships recorded at activation potentials of –80,–110 and –140 mV were linearly extrapolated and the reversal potential was calculated from average of the potentials of the three intersection points of the three linear fits.

## Perforated patch recordings from cMFBs

For perforated-patch recordings from cMFBs, a stock solution was prepared by dissolving the pore-forming antimycotic nystatin in DMSO (25 mg/ml). Immediately before the experiments, the nystatin-stock was added to the intracellular solution at a final concentration of 50 µg/ml. In order to monitor the integrity of the perforated membrane patch, the green-fluorescent dye Atto 488 was added

at a concentration of 50 µM. As nystatin is known to impair the formation of the GΩ seal, the initial ~500 µm of the pipette tip was filled with a perforating agent-free internal solution (tip-filling) before back-filling the pipette shaft with the perforating agent-containing solution. After establishing a GΩ seal, the holding potential was set to –70 mV and the access resistance ($R_a$) was continuously monitored by applying 10 ms long depolarizing pulses to –60 mV at 1 Hz. Recording the voltage-dependent activation of $I_h$ was begun after $R_a$ dropped below 150 MΩ. Because the perforated membrane patch ruptured spontaneously at $R_a$ <50 MΩ, the access resistance was not comparable to standard whole-cell recordings. To exclude the possibility that the right-shift of the $I_h$ activation curve in the perforated configuration (*Figure 7C*) was caused by the comparatively higher $R_a$, the voltage-dependent activation of $I_h$ was measured under normal whole-cell patch-clamp conditions, using pipettes with small openings that resulted in high access resistances ($R_a$ = 119 ± 12 MΩ). However, in these recordings, the midpoint of $I_h$ activation (–105.5 ± 1.4 mV; n = 8) had a tendency to be left-shifted when compared with regular whole-cell recordings with standard patch pipettes ($R_a$ ≈ 30–60 MΩ; $V_{1/2}$ = –103.3 ± 0.8 mV; n = 36; $P_{T-Test}$ = 0.13). The left-shift of the $I_h$ activation curve measured with high access resistances indicates that the right-shift measured with perforated patch recordings might be underestimated because of the higher $R_a$, which would result in an even higher estimate of the endogenous cAMP concentration (*Figure 7D*).

## Analysis of the ZD sensitivity of Na$^+$ currents

Sodium currents (*Figure 1—figure supplement 1*) were isolated using a modified ACSF containing (in mM): NaCl 105, KCl 2.5, NaHCO$_3$ 25, NaH$_2$PO$_4$ 1.25, glucose 20, CaCl$_2$ 2, MgCl$_2$ 1, TEA 20, 4-AP 5 and CdCl$_2$ 0.2. To avoid underestimation of the true size of the presynaptic Na$^+$ currents because of the voltage-drop through the access resistance, we blocked a portion of the Na$^+$ current with 30 nM TTX. Na$^+$ currents were elicited from a holding potential of –80 mV by a 3-ms-long depolarization to 0 mV. Peak amplitudes and half-durations of Na$^+$ currents were measured from leak-subtracted traces.

## Immunoelectron microscopy

Preembedding immunogold labeling was performed as described (*Notomi and Shigemoto, 2004*). Briefly, adult C57Bl/6 mice were anesthetized with sodium pentobarbital (50 mg/kg, i.p.) and perfused transcardially with a fixative containing 4% formaldehyde, 0.05% glutaraldehyde and 15% of a saturated picric acid in 0.1 M phosphate buffer (PB; pH 7.4). Parasagittal sections through the cerebellum were cut at 50 µm, cryoprotected with 30% sucrose, flash frozen in liquid nitrogen and rapidly thawed. Sections were blocked in 10% normal goat serum and 2% bovine serum albumin (BSA) in tris-buffered saline (TBS) for 2 hr at room temperature, incubated in TBS containing 2% BSA and either guinea pig anti-HCN1 or anti-HCN2 antibody (1 µg/ml, *Notomi and Shigemoto, 2004*) for 48 hr at 4°C, and finally reacted with nanogold-conjugated secondary antibody (Nanoprobes, 1:100) for 24 hr at 4°C. Nanogold particles were amplified with the HQ Silver Enhancement kit (Nanoprobes) for 8 min. Sections were treated in 0.5% osmium tetroxide in PB for 40 min and then 1% aqueous uranyl acetate for 30 min at room temperature, dehydrated, and flat embedded in Durcopan resin (Sigma-Aldrich). Ultrathin sections were cut at 70 nm and observed by a transmission electron microscope (Tecnai 12, FEI, Oregon). Sequential images were recorded from the granule cell layer within a few microns of the surface of ultrathin sections at X26,500 using a CCD camera (VELETA, Olympus). For the reconstruction of a half mossy fiber bouton, 36 serial ultrathin sections were used. Sequential images were aligned and stacked using the TrakEM2 program (*Cardona et al., 2012*). For the measurement of density of immunogold particles for HCN2 on this reconstructed profile, 1260 immunogold particles were counted on the mossy fiber bouton membrane area (73.7 µm$^2$), giving a density of 17.1 particles/µm$^2$. Immunogold particles within 30 nm of the bouton membranes were included in the analysis on the basis of the possible distance of the immunogold particle from the epitope (*Matsubara et al., 1996*). The density of non-specific labeling was estimated using the nuclear membrane of a granule cell located adjacent to the reconstructed mossy fiber bouton. We found 40 immunogold particles on the nuclear membrane area of 60.5 µm$^2$ giving a density of 0.66 particles/µm$^2$, which was 3.9% of the HCN2 labeling density on the mossy fiber bouton.

## Hodgkin-Huxley model of axonal HCN channels

Because we did not intend to simulate the cAMP dependence of HCN channel gating explicitly
(*Hummert et al., 2018*), we created two separate models for 0 and 1 mM intracellular cAMP, which
were based on a previously described Hodgkin-Huxley model (*Kole et al., 2006*) with one activation
gate and no inactivation (*Hodgkin and Huxley, 1952*). In short, the activation gate was described by

$$\frac{dm}{dt} = \alpha_m (1 - m) - \beta_m m$$

with

$$\alpha_m(V) = A \; e^{\frac{-(V_m - V_{1/2})}{V_\alpha}}$$

and

$$\beta_m(V) = A \; e^{\frac{(V_m - V_{1/2})}{V_\beta}}$$

The four free parameters, $A$, $V_{1/2}$, $V_\alpha$, and $V_\beta$ were determined by simultaneously fitting $\alpha_m /(\alpha_m + \beta_m)$ to the steady-state activation curve (see *Figure 7B*) and $1/(\alpha_m + \beta_m)$ to the voltage depen-
dence of the time constant of $I_h$ activation and deactivation (*Figure 8B*). The sum of squared errors
was minimized using the FindMinimum routine of Mathematica (version 10; Wolfram Research,
Champaign, IL), with the time constants of activation and deactivation weighed with the inverse of
the square of the maximum value in each of the three datasets (time constant of activation, time con-
stant of deactivation, and steady-state activation curve). The resulting parameters for 0 mM cAMP
were $A$ = 6.907 ms$^{-1}$, $V_{1/2}$ = –102.1 mV, $V_\alpha$ = 18.71 mV, and $V_\beta$ = 21.73 mV. To confirm that the
global minimum was reached, the best-fit parameters were shown to be independent of the starting
values within a plausible range. The 68% confidence interval was calculated as the square roots of
the diagonals of the inverse of the Hessian matrix (*Press et al., 2002*), resulting in ±2.71 ms$^{-1}$, ±16.5
mV, ±17.5 mV, and ±24.3 mV, for $A$, $V_{1/2}$, $V_\alpha$, and $V_\beta$, respectively. We also generated a model for
the corresponding data obtained with 1 mM cAMP in the intracellular solution (cf. *Figure 7B*), result-
ing in $A$ = 7.570 ms$^{-1}$, $V_{1/2}$ = –87.31 mV, $V_\alpha$ = 31.46 mV, and $V_\beta$ = 10.84 mV.

## NEURON model of cMFB

The model of the cMFB consisted of connected cylindrical compartments representing 15 boutons
(length 8 μm and diameter 8 μm) and 15 myelinated axonal compartments (length 35 μm and diame-
ter 0.8 μm; cf. *Palay and Chan-Palay, 1974*; *Figure 9A*). In addition, at one side of this chain, a long
cylinder was added presenting the axon in the white matter (length 150 μm and diameter 1.2 μm).
The specific membrane resistance was 0.9 μF/cm$^2$ (*Gentet et al., 2000*) and the cytoplasmatic resis-
tivity was 120 Ω cm (*Hallermann et al., 2003*). The specific membrane resistance and capacitance of
the axonal compartments were both reduced by a factor of 10, representing myelination.

The ionic membrane conductances were similar to those published by *Ritzau-Jost et al. (2014)*
and were adjusted to reproduce the action potential duration and the maximal firing frequency as
well as the data shown in *Figure 9B–D*. Namely, an axonal Na$^+$ channel (*Schmidt-Hieber and Bis-
chofberger, 2010*) and K$^+$ channel NMODL model (*Hallermann et al., 2012*) was added with a den-
sity of 2000 and 1000 pS/μm$^2$ in the boutons and 0 and 0 pS/μm$^2$ in the axonal compartments,
respectively. The Na$^+$ and K$^+$ reversal potentials were 55 and –97 mV, respectively. To enable the
analysis of the ATP consumption (*Hallermann et al., 2012*), the leak conductance was implemented
as separate Na$^+$ and K$^+$ leak channel models with a conductance of 0.0138 and 0.18 pS/μm$^2$, respec-
tively, in the bouton compartments. In the axonal compartments, both conductances were reduced
by a factor of 10. The above-described Hodgkin-Huxley model of axonal HCN channels (for 0 mM
intracellular cAMP) was added with a density of $g_{HCN}$ = 0.3 and 0.03 pS/μm$^2$ for the bouton and axo-
nal compartments, respectively. For the analysis of the ATP consumption, the $g_{HCN}$ conductance was
separated in a Na$^+$ and a K$^+$ conductance according to $g_{HCN(Na)}$ = (1 – ratio$_{K/Na}$) $g_{HCN}$ and $g_{HCN(K)}$ =
ratio$_{K/Na}$ $g_{HCN}$, where ratio$_{K/Na}$ = ($e_{Na}$ + $e_{HCN}$)/($e_{Na}$ – $e_K$), where $e_{Na}$ and $e_K$ are the Na$^+$ and K$^+$ rever-
sal potential as described above and $e_{HCN}$ is the reversal potential of $I_h$ measured as –23.3 mV (cf.
*Figure 8C*). Assuming a single channel conductance of 1.7 pS for HCN2 channels (*Thon et al.,
2013*), this conductance corresponds to a density of 0.18 HCN channels/μm$^2$, which is much lower

than the estimate from preembedding immunogold labeling (22 particles/$\mu m^2$; *Figure 6*). However, the optimal density of the model critically depends on the geometry of the structure, which was not obtained from the recorded boutons. To obtain the required structural information, including the fenestration of the cMFB (cf. *Figure 6*) and the level of myelination, electron microscopic reconstructions of large volumes of the recorded cMFB and the entire axon would be needed. When we used a $g_{HCN}$, as determined with preembedding immunogold labeling in our model, the model also predicted that $I_h$ critically effects conduction velocity and that the depolarization is the main reason for the velocity to change. In general, these two conclusions of the model were very insensitive to the specific parameters of the model and were, for example, also obtained with additional interleaved cylindrical compartments with high $Na^+$ and $K^+$ channel densities representing nodes of Ranvier, or with a long cylindrical compartment with homogenous channel densities representing an unmyelinated axon. This further supports our finding that HCN channels accelerate conduction velocity independently of the exact parameters of the axon and the degree of myelination (cf. *Figure 1*).

Starting from the model that reproduced the control data, the following four additional models were generated. (1) To simulate ZD application, the HCN HH model was removed. (2) To simulate 8-br-cAMP application, the parameters of the HCN HH model were exchanged with the parameters obtained from the experiments with 1 mM cAMP. (3) To simulate only the depolarization by HCN channels ($V_m$-*model*), the HCN HH channel model was removed and the $K^+$ reversal potential was increased from –97 mV to –90 mV. (4) To simulate only the increase in membrane conductance by HCN channels ($R_m$-*model*), the reversal potential of the HCN HH model was decreased from –23.3 mV to –85.5 mV and the density was increased from 0.3 pS/$\mu m^2$ to 1 pS/$\mu m^2$ in the bouton and from 0.03 pS/$\mu m^2$ to 0.1 pS/$\mu m^2$ in the axon.

All simulations were run with a simulation time interval (*dt*) of <0.2 ms, preceded by a simulation of 1 s with a *dt* of 5 ms to allow equilibration of all conductances. Conduction velocity was calculated from the peak of the action potentials in different boutons of the model. The apparent input resistance was calculated identically to the experimental recordings, that is from the voltage after 300 ms of a –10 pA current injection. IPython (Jupyter Notebooks; *Kluyver et al., 2016*) or Mathematica (Wolfram Research, Champaign, IL) were used to run the NEURON simulations and to visualize and analyze the results (*Hines et al., 2009*).

## Statistics

Statistical analysis was performed using built-in functions of Igor Pro (Wavemetrics, Lake Oswego, OR). The suffix of the P values provided in the legends and the main test indicate the used statistical test. Results were considered significant when P<0.05.

## Code

The NEURON scripts allowing to reproduce the model results will be available at: https://github.com/HallermannLab/2019_HCN (*Hallermann Labratory, 2019*; copy archived at https://github.com/elifesciences-publications/2019_HCN).

## Acknowledgements

We would like to thank Torsten Bullmann for help with implementing the NEURON model in the Python environment and Klaus Nave and the department of Neurogenetics at the Max Planck Institute of Experimental Medicine for scientific support. This work was supported by the German Research Foundation (HA 6386/4–1) to SH.

## Additional information

### Funding

| Funder | Grant reference number | Author |
| --- | --- | --- |
| Deutsche Forschungsgemeinschaft | HA 6386/4-1 | Stefan Hallermann |

The funders had no role in study design, data collection and interpretation, or the decision to submit the work for publication.

## Author contributions
Niklas Byczkowicz, Software, Formal analysis, Validation, Investigation, Visualization, Methodology, Writing—original draft, Writing—review and editing; Abdelmoneim Eshra, Andrea Trevisiol, Investigation, Methodology, Writing—review and editing; Jacqueline Montanaro, Investigation, Visualization; Johannes Hirrlinger, Supervision, Validation, Writing—review and editing; Maarten HP Kole, Conceptualization, Validation, Writing—review and editing; Ryuichi Shigemoto, Supervision, Validation, Investigation, Visualization, Methodology, Writing—review and editing; Stefan Hallermann, Conceptualization, Software, Formal analysis, Supervision, Funding acquisition, Validation, Visualization, Methodology, Writing—original draft, Writing—review and editing

## Author ORCIDs
Niklas Byczkowicz https://orcid.org/0000-0002-6517-287X
Johannes Hirrlinger http://orcid.org/0000-0002-6327-0089
Maarten HP Kole https://orcid.org/0000-0002-3883-5682
Ryuichi Shigemoto http://orcid.org/0000-0001-8761-9444
Stefan Hallermann https://orcid.org/0000-0001-9376-7048

## Ethics
Animal experimentation: All experiments were approved in advance by the Institutional Ethics Committees and animals were treated in accordance with the European (EU Directive 2010/63/EU, Annex IV for animal experiments), National and Leipzig as well as Göttingen University, guidelines.

## Decision letter and Author response
Decision letter https://doi.org/10.7554/eLife.42766.016
Author response https://doi.org/10.7554/eLife.42766.017

# Additional files

## Supplementary files
• Transparent reporting form
DOI: https://doi.org/10.7554/eLife.42766.014

## Data availability
All data generated or analysed during this study are included in the manuscript and supporting files. Scripts to reproduce the model results is available at: https://github.com/HallermannLab/2019_HCN (copy archived at https://github.com/elifesciences-publications/2019_HCN).

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
