## [Decision Letter]

Thank you for submitting your article "Neuromodulation of HCN channels controls action potential conduction velocity in central axons" for consideration by *eLife*. Your article has been reviewed by two peer reviewers, and the evaluation has been overseen by a Reviewing Editor and Ronald Calabrese as the Senior Editor. The following individual involved in the review of your submission has agreed to reveal his identity: Nace Golding (Reviewer #2).

The reviewers have discussed the reviews with one another and the Reviewing Editor has drafted this decision to help you prepare a revised submission.

Summary:

This study examines the contribution of HCN channels to setting conduction velocity in mammalian axons. The work demonstrates that modulation of HCN channels by neuromodulators has the capacity to increase and decrease the rate of action potential propagation.

Essential revisions:

I am glad to say that both reviewers found that the manuscript was interesting and the experiments were largely convincing, although both identified some areas in need of clarification or improvement through additional experimentation. The key issues were as follows:

1) Demonstrating or assessing the biological existence and/or significance of the phenomenon of HCN modulation changing conduction velocity. Reviewer 1 noted that the concentrations of transmitters applied were relatively high and the question was raised about whether endogenous transmitter was likely to generate significant modulation. Reviewer 2 noted the complementary point, namely that it is difficult to assess the functional significance of the tiny (though significant) changes given the filtering properties of granule cells. Because of these combined comments, additional experiments, analysis, and writing are necessary to provide evidence for the biological relevance of the phenomenon. The idea of optogenetic stimulation was brought up but, in the consultation, it was acknowledged that such an experiment might be beyond the scope of practicality. An alternative of measuring conduction velocity with amphetamine-induced release of endogenous monoamines was suggested. So was the idea of measuring activity dependent modulation of Ih with high-frequency trains of stimuli. Those experiments, or equivalent approaches to illustrate that this phenomenon might occur measurably in the brain and influence cerebellar signal processing, would strengthen the manuscript.

2) Clarification of the influence of Ih-dependent depolarization on other ion channels and electrophysiological parameters. Both reviewers pointed out that the HCN-mediated depolarization could influence Na channels, which could in turn influence conduction; the interaction with other channels should be addressed through experiment and/or modeling.

3) Confirmation that the changes reported can be unambiguously ascribed to HCN channels. As you know, in the initial consultation among editors, concerns were raised about lack of occlusion experiments, the imperfect selectivity of ZD, and the absence of antibody controls. This issue should either be addressed experimentally or the conclusions should be appropriately curtailed and discussion of alternatives should be included.

The major points as expressed by the reviewers, with annotations relating to the essential revisions are included below to aid you in your revision.

General assessments:

*Reviewer #1:*

This work by Byczkowicz and colleagues explores the role of HCN channels in axons, focusing on two axonal populations in cerebellum as well as optic nerve preparations. Using a complement of field recording, whole cell recording from boutons, EM, and modeling, they demonstrate that HCN channels bidirectionally control AP propagation speed across the range of preparations. While HCN expression and function has been studied in axons in the past, relatively little is known about how modulation of HCN in axonal areas beyond the initial segment affects excitability. Work here provides strong evidence that such modulation can have an impact on propagation speed and provides detailed mechanistic insight-through empirical and theoretical approaches-into how such modulation occurs. Experiments are well controlled and illustrated beautifully.

*Reviewer #2:*

This study demonstrates a role for hyperpolarization and cyclic nucleotide gated (HCN) cation channels in providing a substrate for modulating the conduction velocity of several types of axons, with a more specific mechanistic focus on the mossy fibers of the cerebellum. The authors identify cAMP as a critical mechanism for the action of at least some neuromodulators, acting through HCN2 subunits. While the action of cAMP on HCN channels is well known, the focus on conduction velocity is new, and the quality of the data is high. The data set also includes a number of impressive, technically challenging experiments, such as perforated patch-clamp recordings from mossy fiber terminals and estimations of cAMP concentration in the terminal. The addition of modeling adds mechanistic clarity to the study. Experiments with three different axon types shows that the effects of cAMP and Ih on conduction velocity may be widespread in CNS axons. I do have some suggestions for improvements, detailed below.

Major comments (combined, with indications as to which essential revision is being spelled out in more detail):

1) Related to Essential Revision 1: While the data are unequivocal that such modulation can occur, I'm left wondering if this is indeed a mechanism employed to regulation information transfer in the cerebellum or other structures. The main concern involves Figure 2. Here, application of various neuromodulations demonstrate the capacity for modulation by endogenous systems, but they do little to show which of these systems are involved in such regulation. The concentrations used are so high that off target effects are a strong possibility (e.g., norepinephrine can engage dopamine receptors at high concentrations, Sanchez-Soto et al., 2016). It is critical to demonstrate that these effects are blocked by specific antagonists to establish that all of these systems are indeed in play. Furthermore, replicating these results using endogenous sources is a critical step to determine whether such modulation occurs in vivo. This would ideally be done with optogenetic activation of monoaminergic sources. (Ed's note: Note that optogenetic expts are not required for the revision, but this sentence is included since it was in the original review.) Less ideal, but still towards the point, would be to apply amphetamine to reverse monoamine transport. Either approach should be paired with appropriate antagonist controls.

2) Related to Essential Revision 1: The functional significance for the effects of Ih modulation on conduction velocity is somewhat vague as it is presented. The differences in conduction velocity are small, and on a sub-millisecond time scale. While the authors cite several papers supporting the importance of timing on this scale, the discussion lumps systems with very different circuit structures and computational demands. What is the evidence for the significance of submillisecond timing in the cerebellum? The mossy fiber input is ultimately low-pass filtered by granule cells with very slow membrane time constants so this requires some comment.

3) Related to Essential Revision 2: A second question relates to the modeling: The modeling explores two competing hypotheses that could account for the observed effects of HCN modulation, relating to changes in membrane potential and leak, and show that membrane potential and its relation to spike threshold is sufficient to explain the observed effects. This can be explained well by the cell being closer to threshold. A counter-argument would be that depolarization decreases NaV availability, but the authors mention that this is not a confound. The authors would be well served to explore this aspect in more detail to see how robust this result is across a range of conditions. First, it would be report relative NaV availability in each model. Second, it would be great to see if the relative changes in conduction velocity hold if the model starts at other membrane potentials (particularly more depolarized).

4) Related to Essential Revisions 1 and 2, as stated by the other reviewer: The modulation of Ih is presented as a way to dynamically control conduction velocities according to changes in neuromodulatory state. Given the presence of voltage-gated calcium channels in some axons and nodes of Ranvier, is it possible that the cAMP-mediated modulation of Ih could also serve a homeostatic role, bringing Vrest closer to threshold and offsetting the inactivation of Na channels under conditions of high action potential firing rates? This would be straightforward to test with high frequency trains of axonal stimuli in experiments and perhaps with the model.

[Editors' note: further revisions were requested prior to acceptance, as described below.]

Thank you for resubmitting your work entitled "HCN channel-mediated neuromodulation can control action potential velocity and fidelity in central axons" for further consideration at *eLife*. Your revised article has been favorably evaluated by Ronald Calabrese (Senior Editor), a Reviewing Editor, and two reviewers.

The manuscript has been substantially improved but there are a few remaining issues that we would like you to address before acceptance, as outlined below:

Essential revisions:

Both reviewers agree that the manuscript has been improved by the new experiments. Because of some remaining questions regarding neuromodulation and possible consequences of action potential failure, both reviewers also agree that additional specific discussion of these issues is warranted. We are including the specific language of both reviewers below, because they have expressed the points clearly and can guide you in those textual revisions.

*Reviewer #1:*

The authors have done a good job addressing reviewer concerns, and I am very supportive of publication with a few additional text/formatting updates. Of note, the new recordings described in Figure 5 are impressive and highlight an interesting aspect of HCN function in axons, contributing to reliability during high frequency trains. I was a bit disappointed that neuromodulatory antagonist experiments were not performed. Instead, experiments were done with ZD and agonist application. This isn't as clean, given the issues related to ZD specificity, but nevertheless support the notion that HCN channels are the ultimate effector of the neuromodulatory systems engaged. In light of this, I'd request that the discussion relating to how neuromodulators could regulate axonal conduction be expanded to include any details known on neuromodulatory mechanisms previously reported in these axons, and a bit more on the possibility of off-target interactions with these concentrations. Also, the data in Figure 5 deserve a bit of discussion. AP failures at high frequency may be more consequential than these small changes in timing.

Reviewer #2:

This study explores the role of HCN channels in regulating action potential conduction velocity in mossy fibers of the cerebellum and other axons as well. In the revision, the authors have provided a significant number of new experiments that add value to the manuscript. The terminal recordings combined with the modeling together convincingly demonstrate a significant impact of HCN channel activity not only on AP propagation velocity, but also on presynaptic spike frequency and reliability. The modeling better explores the impact of HCN channels on Na channel availability. The magnitude of HCN channel modulation on conduction velocity in vivo remains to be determined, but with the addition of the new experiments and modeling, this study provides a sophisticated examination of the interplay between HCN channels and voltage-gated Na channels in presynaptic axons. As before, the quality of the data is excellent. My only remaining concern revolves around the added occlusion experiments in Figure 3, as they relate to the neuromodulation experiments in Figure 2. In the presence of ZD7288 (30 µM), there are changes in conduction velocity of ~3-4%. This range is comparable to the neuromodulator-induced changes in velocity seen in the absence of ZD in Figure 2, with the exception of adenosine. In this regard, it would seem that these experiments are only partially successful in demonstrating specificity of neuromodulatory action on HCN channels. I think this issue needs to be reflected in the writing. What is explanation for the remaining ZD-insensitive neuromodulation?

---

## [Author Response]

Essential revisions:I am glad to say that both reviewers found that the manuscript was interesting and the experiments were largely convincing, although both identified some areas in need of clarification or improvement through additional experimentation. The key issues were as follows:1) Demonstrating or assessing the biological existence and/or significance of the phenomenon of HCN modulation changing conduction velocity. Reviewer 1 noted that the concentrations of transmitters applied were relatively high and the question was raised about whether endogenous transmitter was likely to generate significant modulation. Reviewer 2 noted the complementary point, namely that it is difficult to assess the functional significance of the tiny (though significant) changes given the filtering properties of granule cells. Because of these combined comments, additional experiments, analysis, and writing are necessary to provide evidence for the biological relevance of the phenomenon. The idea of optogenetic stimulation was brought up but, in the consultation, it was acknowledged that such an experiment might be beyond the scope of practicality. An alternative of measuring conduction velocity with amphetamine-induced release of endogenous monoamines was suggested. So was the idea of measuring activity dependent modulation of Ih with high-frequency trains of stimuli. Those experiments, or equivalent approaches to illustrate that this phenomenon might occur measurably in the brain and influence cerebellar signal processing, would strengthen the manuscript.

We agree that, to unequivocally demonstrate the biological existence of neuromodulation of action potential propagation velocity, in vivo experiments are best-suited. However, we also feel that this is beyond the scope of this manuscript. To address this concern, we have now revised the manuscript by curtailing the conclusion from HCN channels *do* modulate conduction velocity to HCN channels *can* modulate conduction velocity (e.g., by changing the title to “HCN channel-mediated neuromodulation can control action potential velocity and fidelity in central axons”, in the Abstract “is modulated” to “can be modulated”, and in the Discussion: “However, our data demonstrate that under certain conditions an active control of conduction velocity could occur in the vertebrate CNS via the cAMP-HCN pathway”).

In the revised manuscript, we followed the suggestion of the reviewing editor and the reviewers by investigating high-frequency trains of action potentials. To this end, we performed a set of new experiments. We made additional whole-cell recordings from cerebellar mossy fiber boutons, loaded the bouton and the adjacent axon with a fluorescent dye, positioned a second extracellular stimulation pipette (filled with another dye) through two-photon fluorescent-guided microscopy close to the remote part of the axon, elicited high-frequency trains of action potentials with the stimulation pipette, and recorded the arriving action potentials in whole-cell current-clamp at the mossy fiber bouton (see new Figure 5 and Materials and methods, subsection “Recordings from cMFBs”). These experiments allowed unequivocal determination of the maximal failure-free frequency of individual axons as well as changes of conduction velocity during high-frequency trains of action potentials. The results revealed a strong reduction of the failure-free frequency from about ~800 to ~400 Hz upon blocking of HCN channels with ZD7288 (n = 20 and 10, respectively; P_T-Test_ = 0.0002). Furthermore, the conduction velocity during the train of action potentials decreased by 20% in the presence of ZD7288 but increased by 5% in control recordings. Thus, in addition to the ~5% change in baseline conduction velocity reported in the initial version of the manuscript, the lack of HCN channels further decelerates action potential propagation during high-frequency trains of action potentials up to ~25% and decreases the maximal failure-free frequency by a factor of two. Our additional experiments therefore further support the idea that HCN channels have an important role in controlling axonal action potential propagation.

2) Clarification of the influence of Ih-dependent depolarization on other ion channels and electrophysiological parameters. Both reviewers pointed out that the HCN-mediated depolarization could influence Na channels, which could in turn influence conduction; the interaction with other channels should be addressed through experiment and/or modeling.

To address the contribution of the inactivation of voltage-gated sodium (Na_V_) channels upon depolarization, we have now systematically varied the resting membrane potential of our model by changing the potassium reversal potential (Figure 10—figure supplement 1 of the revised manuscript). Increasing the resting membrane potential from –90 to –45 mV (by increasing the potassium reversal potential from –120 to –50 mV) increased the proportion of inactivated Na_V_ channels from 10 to 80%. Despite increasing inactivation of Na_V_ channels the conduction velocity continued to increase. These results are described in subsection “Mechanism of conduction velocity-control and metabolic costs of HCN channels” and discussed as follows: “However, our modelling results showed that increasing the membrane potential from –90 to –60 mV increased the conduction velocity despite significantly decreasing Na^+^ channels availability (Figure 10—figure supplement 1). Interestingly, these findings are in agreement with the nonlinear cable theory predicting that the difference between the resting membrane potential and the firing threshold is a critical parameter for action potential conduction velocity (see, e.g., Figure 12.25 in Jack et al., 1983, for increasing velocity with increasing safety factor, i.e. decreasing excitation threshold *V*_B_). Intuitively, the HCN channel mediated acceleration of conduction velocity can be understood as follows; in a more depolarized axon, Na_V_ mediated current influx in one axonal location will depolarize neighboring locations faster above the threshold. In our model, this effect outweighs the disadvantage of the increased steady-state inactivation of Na^+^ channels up to a membrane potential of about –65 mV and a Na_V_ availability of 50%. The exact values, above which Na_V_ availability limits conduction velocity, critically depend on assumptions of the model, such as the voltage-dependence of inactivation and the density of Na_V_ channel.”

3) Confirmation that the changes reported can be unambiguously ascribed to HCN channels. As you know, in the initial consultation among editors, concerns were raised about lack of occlusion experiments, the imperfect selectivity of ZD, and the absence of antibody controls. This issue should either be addressed experimentally or the conclusions should be appropriately curtailed and discussion of alternatives should be included.

We thank the reviewers for bringing this point to our attention. In the revised manuscript, we have performed new experiments, in which we applied neuromodulators in the presence of ZD 7288. The neuromodulators (cAMP, adenosine, and norepinephrine) had no impact on conduction velocity in the presence of ZD 7288. The results from these occlusion experiments further support our conclusion that HCN channels mediate the effect of neuromodulators on conduction velocity. See new Figure 3 and subsection “Neuromodulation of conduction velocity is mediated by HCN channels”.

The major points as expressed by the reviewers, with annotations relating to the essential revisions are included below to aid you in your revision.Major comments (combined, with indications as to which essential revision is being spelled out in more detail):1) Related to Essential Revision 1: While the data are unequivocal that such modulation can occur, I'm left wondering if this is indeed a mechanism employed to regulation information transfer in the cerebellum or other structures. The main concern involves Figure 2. Here, application of various neuromodulations demonstrate the capacity for modulation by endogenous systems, but they do little to show which of these systems are involved in such regulation. The concentrations used are so high that off target effects are a strong possibility (e.g., norepinephrine can engage dopamine receptors at high concentrations, Sanchez-Soto et al., 2016). It is critical to demonstrate that these effects are blocked by specific antagonists to establish that all of these systems are indeed in play. Furthermore, replicating these results using endogenous sources is a critical step to determine whether such modulation occurs in vivo. This would ideally be done with optogenetic activation of monoaminergic sources. (Ed's note: Note that optogenetic expts are not required for the revision, but this sentence is included since it was in the original review.) Less ideal, but still towards the point, would be to apply amphetamine to reverse monoamine transport. Either approach should be paired with appropriate antagonist controls.

Please see detailed response to point 1 above. In the revised manuscript, we also mentioned potential off-target effects: “Although we used rather high concentrations of the agonists and off-target effects cannot be excluded (e.g., NE activating dopamine receptors; Sánchez-Soto et al., 2016), these data nevertheless indicate that…” Furthermore, our new experiments (Figure 3) show that the effect of the neuromodulators is mediated by HCN channels.

2) Related to Essential Revision 1: The functional significance for the effects of Ih modulation on conduction velocity is somewhat vague as it is presented. The differences in conduction velocity are small, and on a sub-millisecond time scale. While the authors cite several papers supporting the importance of timing on this scale, the discussion lumps systems with very different circuit structures and computational demands. What is the evidence for the significance of submillisecond timing in the cerebellum? The mossy fiber input is ultimately low-pass filtered by granule cells with very slow membrane time constants so this requires some comment.

We have now more carefully discussed the significance of submillisecond timing in the cerebellum and added the paragraph: “The cerebellum is involved in the accurate control of muscle contraction with a temporal precision of 1-100 ms (Hore et al., 1991). Submillisecond correlations in spike timing occuring between neighboring Purkinje cells have been noted previously (reviewed in Isope et al., 2002; Person and Raman, 2012). Furthermore, submillisecond precision of the mossy/parallel fiber input are critical for processing in the cerebellar circuits (Braitenberg et al., 1997; Heck et al., 2001). Together, the here-described changes in action potential conduction velocity in mossy and parallel fibers (Figures 1-3) may thus play an important role in cerebellar computation. Furthermore, the observed impairment in high-frequency firing without HCN channels (Figure 5) is expected to negatively impact such functions (Delvendahl and Hallermann, 2016).”

Regarding the membrane time constant of granule cells, we agree that filtering at the granule cell soma (and the Purkinje cell dendrites) will occur but we would like to point out that the membrane time constants of granule cells strongly changes during development (Cathala et al., 2003, J Neurosci 23:6074-85) and that the spiking delay and precision critically depend on other parameters such as the distance to firing threshold.

3) Related to Essential Revision 2: A second question relates to the modeling: The modeling explores two competing hypotheses that could account for the observed effects of HCN modulation, relating to changes in membrane potential and leak, and show that membrane potential and its relation to spike threshold is sufficient to explain the observed effects. This can be explained well by the cell being closer to threshold. A counter-argument would be that depolarization decreases NaV availability, but the authors mention that this is not a confound. The authors would be well served to explore this aspect in more detail to see how robust this result is across a range of conditions. First, it would be report relative NaV availability in each model. Second, it would be great to see if the relative changes in conduction velocity hold if the model starts at other membrane potentials (particularly more depolarized).

We thank the reviewer for this suggestion. Our systematic analysis of the interplay between resting membrane potential and Na_V_ channel availability is explained above (general point 2). As suggested, we now provide the Na_V_ availability in each model: “For the control model, the Na_V_ inactivation was 12%. For the model reproducing the experiments with ZD7288 and 1 mM intracellular cAMP, the Na_V_ inactivation was 6 and 17%, respectively. Because the steady-state Na_V_ availability depends mostly on the resting membrane potential, the relation between resting membrane potential and Na_V_ availability was identical for all three models (data not shown).”

4) Related to Essential Revisions 1 and 2, as stated by the other reviewer: The modulation of Ih is presented as a way to dynamically control conduction velocities according to changes in neuromodulatory state. Given the presence of voltage-gated calcium channels in some axons and nodes of Ranvier, is it possible that the cAMP-mediated modulation of Ih could also serve a homeostatic role, bringing Vrest closer to threshold and offsetting the inactivation of Na channels under conditions of high action potential firing rates? This would be straightforward to test with high frequency trains of axonal stimuli in experiments and perhaps with the model.

We would like to thank the reviewer for this insightful comment. As explained in more detail above (general point 1) we have performed new experiments to analyze the conduction velocity during high-frequency trains of action potentials (see new Figure 5D-G). We are thankful for providing the exciting idea of calcium controlling the cAMP/HCN pathway. A mechanistic analysis of the interplay between Ca^2+^and the cAMP/HCN pathway is beyond the scope of this manuscript, but we have added the following sentences: “Interestingly, Ca^2+^ entering through axonal voltage-gated Ca^2+^ channels (Bender et al., 2010) could interact with the cAMP pathway by activating or inhibiting different subtypes of adenylyl cyclase and phosphodiesterase (Bruce et al., 2003). Thereby, HCN could also serve a homeostatic role, by bringing the resting membrane potential closer to threshold and offsetting the inactivation of Na_V_ channels under conditions of high-frequency action potential firing.”

[Editors' note: further revisions were requested prior to acceptance, as described below.]

Essential revisions:Both reviewers agree that the manuscript has been improved by the new experiments. Because of some remaining questions regarding neuromodulation and possible consequences of action potential failure, both reviewers also agree that additional specific discussion of these issues is warranted. We are including the specific language of both reviewers below, because they have expressed the points clearly and can guide you in those textual revisions.

Reviewer #1:

The authors have done a good job addressing reviewer concerns, and I am very supportive of publication with a few additional text/formatting updates. Of note, the new recordings described in Figure 5 are impressive and highlight an interesting aspect of HCN function in axons, contributing to reliability during high frequency trains. I was a bit disappointed that neuromodulatory antagonist experiments were not performed. Instead, experiments were done with ZD and agonist application. This isn't as clean, given the issues related to ZD specificity, but nevertheless support the notion that HCN channels are the ultimate effector of the neuromodulatory systems engaged. In light of this, I'd request that the discussion relating to how neuromodulators could regulate axonal conduction be expanded to include any details known on neuromodulatory mechanisms previously reported in these axons, and a bit more on the possibility of off-target interactions with these concentrations. Also, the data in Figure 5 deserve a bit of discussion. AP failures at high frequency may be more consequential than these small changes in timing.

We thank the reviewer for the positive evaluation of our manuscript. We have extended the discussion and added a paragraph (subsection “Mechanism and metabolic costs of HCN channel mediated control of action propagation” paragraph two) explaining the results of Figure 4 of the revised manuscript.

Reviewer #2:

This study explores the role of HCN channels in regulating action potential conduction velocity in mossy fibers of the cerebellum and other axons as well. In the revision, the authors have provided a significant number of new experiments that add value to the manuscript. The terminal recordings combined with the modeling together convincingly demonstrate a significant impact of HCN channel activity not only on AP propagation velocity, but also on presynaptic spike frequency and reliability. The modeling better explores the impact of HCN channels on Na channel availability. The magnitude of HCN channel modulation on conduction velocity in vivo remains to be determined, but with the addition of the new experiments and modeling, this study provides a sophisticated examination of the interplay between HCN channels and voltage-gated Na channels in presynaptic axons. As before, the quality of the data is excellent. My only remaining concern revolves around the added occlusion experiments in Figure 3, as they relate to the neuromodulation experiments in Figure 2. In the presence of ZD7288 (30 µM), there are changes in conduction velocity of ~3-4%. This range is comparable to the neuromodulator-induced changes in velocity seen in the absence of ZD in Figure 2, with the exception of adenosine. In this regard, it would seem that these experiments are only partially successful in demonstrating specificity of neuromodulatory action on HCN channels. I think this issue needs to be reflected in the writing. What is explanation for the remaining ZD-insensitive neuromodulation?

We would like to thank the reviewer for these positive comments. We believe that the remaining concern is based on a misunderstanding. The reviewer argues that the occlusion experiments are only partially successful in demonstrating specificity of neuromodulatory action on HCN channels because the conduction velocity changed in the presence of 30 µM ZD7288 by ~3-4% (Figure 3; now Figure 2F and G in the revised manuscript), which is similar to the neuromodulator-induced changes in velocity in the absence of ZD (Figure 2; now Figure 2C-D). However, a direct comparison of the change in conduction velocity between Figure 2 and 3 of the previous manuscript is not informative. To better explain these results, we used horizontal lines to illustrate the time of application of the substances in Figure 2C and F and added the following sentences in the Materials and methods section of the revised manuscript:

“To investigate the contribution of HCN channels on the neuromodulation of the conduction velocity (Figure 2F-G), cerebellar slices were pre-incubated for 20 min at room temperature with 30 µM ZD7288 and then transferred to the recording chamber (35 °C) with continued application of 30 µM ZD7288. 5 min after the beginning of the recordings (t = 0 min), the neuromodulators were applied. Note that the control data without application of neuromodulators (“only ZD7288” in Figure 2F and G) show a larger decrease in conduction velocity during the 20-min recording period compared with the control data in Figure 2C-E. The difference in the control data is most likely due to the slow action of ZD7288 (cf. Figure 1), which further decreases the conduction velocity during the recordings in addition to the run-down observed in control recordings without ZD7288. Therefore, the change in conduction velocity induced by the neuromodulators with and without pre-incubation of ZD7288 (Figure 2F-G and 2C-E, respectively) cannot be compared directly but have to be compared with the corresponding controls.”